# Biodiversity conservation in cities: Defining habitat analogues for plant species of conservation interest

**M. Itani**[1,2], **M. Al Zein**[3], **N. Nasralla**[2], **S. N. Talhouk**[1,2]*

**1** Department of Landscape Design and Ecosystem Management, Faculty of Agricultural and Food Sciences, American University of Beirut, Beirut, Lebanon, **2** Nature Conservation Center, American University of Beirut, Beirut, Lebanon, **3** Department of Biology, Faculty of Arts and Sciences, American University of Beirut, Beirut, Lebanon

* ntsalma@aub.edu.lb

**Data Availability Statement:** All the raw data has been uploaded under supplemental information.

**Funding:** The author(s) received no specific funding for this work.

## Abstract

Urban green spaces, both unmanaged and managed, include novel ecosystems that may be suitable habitat analogues for native plant species of conservation interest. The objective of this study was to define habitat analogues in the Mediterranean city of Beirut for *Matthiola crassifolia*, a Lebanese steno-endemic only present in urban habitats. We adopted a step-wise method that integrates two vegetation assessments, floristics, and life form. We placed seventy-eight quadrats (1m x 1m) in 12 study sites following a deliberate biased approach to capture habitat diversity. In every quadrat, we performed taxonomic identification and recorded life forms of each species. We pooled species that shared the same life form and estimated area cover for each life form accordingly. We performed TWINSPAN analyses on both floristic and life form data, then combined these findings to generate a description of habitat analogues suitable for *M. crassifolia*. TWINSPAN analysis of floristic data clustered the 78 quadrats under 17 quadrat groups, while life form data assembled the quadrats under 11 quadrat groups. The integration of floristic and life-form classification results into one matrix generated 30 quadrat groups, 8, which were highly favorable to *M. crassifolia*, and 12, which excluded it. The stepwise method unveiled similarities between vegetation assemblages, which appeared distinct due to the high presence of ruderals. We found that habitat analogues favorable to *M. crassifolia* include green spaces dominated by palms, low-lying succulents, or shrubs with scale-like leaves. In contrast, areas dominated by turf grass, canopy trees, or vegetation that produces significant litter were not favorable to *M. crassifolia*'s persistence. Based on these findings, we generated a plant palette of life forms which guides designs of urban habitats favorable to *M. crassifolia*.

### Synthesis and applications

The stepwise method was useful in producing informative plant lists and assemblages for planting designs and landscape management; it generated a plant selection palette that is not restrictive and does not enforce a native only policy. It also offered a wide range of potential habitat analogues for *M. crassifolia*.

**Competing interests:** The authors have declared
that no competing interests exist.

## Introduction

Novel ecosystems are human-modified ecosystems that have been irreversibly altered by intense impacts on abiotic conditions or biotic composition [1, 2, 3]. Novel ecosystems include urban green spaces that emerge mostly after built structures have replaced previously existing ecosystems. As such they include non native vegetation assemblages, consisting of native, spontaneous, naturalized, and invasive species [4]. Urban green spaces are sometimes abandoned after human disturbances or continue to experience disturbance regimes and consequently contain a range of both early- and late-succession vegetation.

Both unmanaged and managed green space can potentially contribute to urban biodiversity conservation. When unmanaged, urban green spaces are referred to as Informal Green Spaces (IGS) and can potentially contribute to urban biodiversity conservation [4]. IGS can provide valuable habitats [5, 6, 7, 8, 9, 10], and occasionally serve as a substitute for natural habitats [11, 12]. Certain cities are important for the conservation of threatened species [13]. Urban green spaces in Mediterranean cities, for instance, where plant diversity and endemism are high, offer a prospective of refuges to plant species regardless of whether the urban green spaces are semi-natural or anthropogenic [14]. However, despite the persistence of endangered species in cities [15], there are only few reported case studies of cities hosting viable populations of rare or endangered species, and thus directly contributing to conservation efforts [16].

Urban habitats tend to favor the persistence of plant species with particular trait combinations that appear well suited to the conditions [17]. Furthermore, certain plant functional traits tend to increase in response to urbanisation, while other traits have mixed responses [18]. For example, it has been shown that urbanized grid cells favor wind pollinated plants, plants with scleromorphic leaves, or plant seeds dispersed by animals, while other grid cells favor insect pollinated plants, plants with hygromorphic leaves, or plant seeds dispersed by wind [19]. Also, acidophiles may have disappeared in urban areas [20]. Identifying predictable relationships between plant traits and environmental conditions or disturbances is a promising approach for understanding how plant communities change in response to human land-use modification [21].

Many plant species can be found more or less regularly in various city habitats; the region of a habitat associated with a particular ecological community. For example, classification of urban habitat types inside the city of Berlin has revealed 19 habitats particularly worthy of protection and these were nominated as legally protected [22]. However, the classification of habitat types inside cities requires standard habitat classification systems which have not been developed in all countries, at least not in Lebanon [23]. Furthermore, the nomination of such habitat types becomes challenging when urban habitats are privately owned, as is the case of most informal green spaces in Beirut [24]. Another challenge is the protection of such habitats in cities like Beirut where law enforcement is weak and is unlikely to deter against infringement [25].

More relevant to cities like Beirut are urban biodiversity strategies that proposed to transform urban habitats into habitats suitable for native plant conservation [26]. One example of urban biodiversity strategy is the use of species-rich herbaceous communities to promote biodiversity in cities [27]. Another strategy, referred to as reconciliation ecology, proposes the conversion of spaces assigned to human activities into spaces that support the persistence of native species [28]. Identifying habitat analogues in this case is essential to guide reconciliation ecology strategy in cities [29]. If appropriate conservation targets are set, habitat analogues may dilute the distinction between disturbed and non-disturbed habitats as favorable sites for plant conservation [30, 31].

There are various methods that describe vegetation based on species identity and abundance, species functional traits, structural characteristics, or degree of naturalness. Floristic surveys are one of two main vegetation description methods used to collect data on native species of conservation interest, and to generate community classification schemes and structure patterns which vary predictably in response to external factors such as environmental stress and disturbance [32]. The floristic method uses taxonomic identification and species abundance to describe vegetation. From a floristics perspective, plant species found in an area are unique and capable of coexisting as distinct, recognizable units that are repeated regularly in response to biotic and environmental variations [33, 34, 35, 36, 37].

The other main vegetation description method, physiognomy, is frequently used to describe vegetation according to external morphology, life form, stratification, and size of each species [38, 39, 40, 41, 42, 43, 44, 45, 46, 47].

EcoVeg is a recent method that combines floristics and physiognomy, in addition to ecological descriptors, and that applies different rationales depending on whether the vegetation is natural or cultural [48]. Combining both approaches may be necessary to generate informative data from sites subjected to different disturbance conditions. The application of floristics in urban habitats may present a challenge when interpreting the data, since many studies reported an over-representation of ruderal species and high taxonomic diversity between relatively close sites [49, 50, 51, 52, 53, 54, 55, 56, 57, 58, 59]. In contrast, the application of physiognomic and structural vegetation description, may be more useful in urban habitats [18, 60] as the data informs about predominance of life strategies adopted by different life forms, and the method is applicable in highly modified sites, and at both, macro- and micro-climate conditions [61, 62].

A widely used functional type or physiognomic vegetation description system is the life form classification of Raunkiær [39]. Although physiognomic and structural vegetation description methodologies were developed to describe vegetation over large areas, these methods have been considered as potentially more useful tools than floristics in highly modified sites because they better reflect life strategies, for example, the ruderal strategy, encompassed by certain life forms [61]. Down (1973) resorted to life-form in studying reclamation of spoil heaps [62].

There is consensus that physiognomic and physiological characteristics of plants, including species life-history strategies and population biology, are also important descriptors of vegetation communities [63, 64, 65, 66, 67, 68, 69, 70]. Plant communities were shown to be important indicators to determine suitable habitats for rare species [71], and for ecologically and economically important species [72]. Some of these studies, however, deliberately exclude disturbed areas from sampling [72].

One aspect of urban vegetation that might challenge classification is the abundance of ruderal plant species which, benefit from the absence of interspecific competition that normally occurs in later successional stages, and colonize bare and disturbed land [61]. By spreading from nearby semi-natural vegetation, ruderals contribute to high variability in urban plant diversity, even between close sites, limiting the value of vegetation classification using floristic methods [32]. Some of these ruderal species may be distantly related to agricultural weeds and others to plant species found across transportation networks [73]. Ruderals are also populating green walls in cities [74]. The overrepresentation of ruderals and the haphazard management of green spaces in Beirut make vegetation classification difficult due to the small scale of associated biotopes and abundance of structured biotope complexes.

The success of plant conservation strategies is highly influenced by perceptions and social preferences which should be taken into consideration in addition to field assessment challenges in cities. For example, studies have shown that spontaneous 'unmanaged' vegetation

may not appeal to residents as aesthetically pleasing nor is it perceived as acceptable 'urban nature' by decision-makers [75, 76, 77]. This is further complicated by the fact that plant selection and management, is driven by landscape architects and landscape contractors who have limited experience with native species, and do not have clear guidelines on how to contribute to biodiversity conservation in cities [78, 79].

While several studies show how vegetation description using floristic assessments in urban areas is limited by over-representation of ruderal species, abundance of exotic plants, and a high taxonomic diversity between relatively close sites [49, 50, 51, 52, 73, 54, 55, 58, 80, 81], other studies suggest that descriptions of functional types, such as life form, may permit ecological comparisons among areas of similar ecology on a more general scale than would be possible when using a taxonomic approach [82, 83, 84]. For instance, structural and adaptation characteristics of beach and dune vegetation were found similar, even if their taxonomic spectrum differed [85]. Furthermore, life-form, among other descriptions of functional types, were associated with plant responses to environmental change, to plant competitive strength, and to plant effects on biogeochemical cycles and disturbance regimes [86]. Recently, life form and life history were found to be stronger predictors of underlying population processes than native status [87]. The first meta-analysis on intra-urban biodiversity variation worldwide showed that patch area and corridors have the strongest positive effects on biodiversity, and that vegetation structure, local scale, biotic factors, and management habitat variables, are significantly more important than landscape scale, abiotic factors, or design related variables [88].

The objective of this study was to define urban habitat analogues for a plant species of conservation interest, *Matthiola crassifolia*, which has persisted in varying abundance in the Mediterranean city of Beirut. *M. crassifolia* is a rare Lebanese steno-endemic, it is only present in urban habitats, and its largest population in Beirut is decreasing. As the natural habitat of *M. crassifolia* is described as coastal area rocks [89], we hypothesize that the expected habitat analogues for the target species will include urban green space typologies that include significant percent of bareground, minimal presence of plant litter, and vegetation that does not significantly produce shade.

## Materials and methods

### Study location

Located along the Eastern shores of the Mediterranean, Lebanon is a predominantly mountainous country consisting of five geomorphological regions namely, a narrow coast along the length of the country, two mountain ranges that run parallel to the sea, and a fertile high plain that separates the two mountain chains. Lebanon possesses botanical elements from temperate, arid and subtropical biomes.

### Species of conservation interest and its distribution

There are four *Matthiola* species recorded in Lebanon, two of which are either national or regional endemics. The Species-Group Ovatifolia is represented by the regional endemic *Matthiola damascena* Boiss. The Species-Group Longipetala is represented by *Matthiola tricuspidata* and *Matthiola longipetala*. Species-Group Iincana is represented by the national endemic *Matthiola crassifolia* Boiss. & Gaill which is restricted to a few locations along the highly urbanized Lebanese coast and is the subject of this study. *M. crassifolia* is a taxon of conservation interest as the species is recognized as an endemic of Lebanon. However, Gowler (1998) has questioned the taxonomic status of the species proposing that it should be considered

subspecies of *Matthiola sinuata* [90]. Even if future molecular analyses support this preference, the taxon will remain an endemic of Lebanon yet at the intra-specific level.

The most comprehensive record of the distribution and status of *M. crassifolia* prior to this study was by Rteil (2002) who performed a systematic survey of the Lebanese coast and recorded the presence of the species in three out of five previously reported sites, Beirut, Ras Beirut and Byblos [91]. In this study, Ras Beirut and Beirut were considered a single locality. Subsequent field investigations by added Sidon, Khaldeh and Amchit as localities for *M. crassifolia* [89]. Our field survey to all reported localities confirmed the extinction of *M. crassifolia* in Sidon and its continued presence in Khaldeh, Beirut, Amchit and Byblos [92].

## Study area

Beirut (33.8869˚ N, 35.5131˚ E), the capital of the Republic of Lebanon, is located on the eastern coast of the Mediterranean. Archeological evidence shows that humans have continuously occupied Beirut for the last 5000 years [93, 94]. Today, the city of Beirut has one of the highest urban densities in the Middle East with an area roughly over 20 km$^2$, population density is estimated at 21,000 people per sq. km [95, 96]. The topography of the city includes two hills, Achrafieh (100 m asl) and Mousseitbeh (80 m asl) [97]. Paul Mouterde, who conducted floristic studies in Beirut in the 20th century, reported 1200 floral species including native and non-native species [98].

Our study site, Beirut, is defined by a 6 km long and 2 km wide cape [99]. Today, this area consists of densely populated neighborhoods interspersed with managed landscapes and zones with spontaneous naturalized vegetation occurring within geographically adjacent lots. Recent floristic studies of semi natural areas of Beirut revealed low community similarity, patchy species distribution, and predominance of habitat non-specific species [81]. Green spaces in the southern part of the promontory of Beirut fall under two broad categories; managed landscapes, dominated by exotic ornamental species planted in raised beds with reconstructed soil, and spontaneous landscapes where spontaneous floral communities survive along with casual non-native species, in coastal cliffs, along the rocky water front, and in un-built/abandoned lots [100].

The study location, particularly the southern part of the promontory of Beirut, can be considered a type III city that is likely to be carrying an extinction debt because extensive landscape transformations occurred after initial floristic surveys [80]. Although the expansion of the city started in 1840, the city passed through five stages of transformation when the southern part of the promontory consisted mostly of semi-natural areas until 1943 [101]. The earliest botanical studies in the region took place in the mid-1800 [102] and continued giving considerable focus to Beirut and its environments till the 1930s [103; 104].

Concurrent with early botanical studies of semi natural areas in Beirut, since 1840, Beirut has passed through five phases of transformation which extensively altered its landscape (Fig 1) [101]. Today, Beirut still harbors significant remnant native vegetation, especially the southern side of the promotary where urban expansion took place after 1970.

## Field data collection

Baseline data collection was initiated in 2012, three years before the start of the study, to ensure comprehensive coverage of all informal green space locations. During 2012, all informal green spaces in the study location were surveyed to locate all spaces where *M. crassifolia* was present. In subsequent years (2013 and 2014), during flowering season of the target species, annual visits were made to all identified green spaces, regardless of whether the species was present or not. In 2015, sites were selected according to management intensity (high or low management

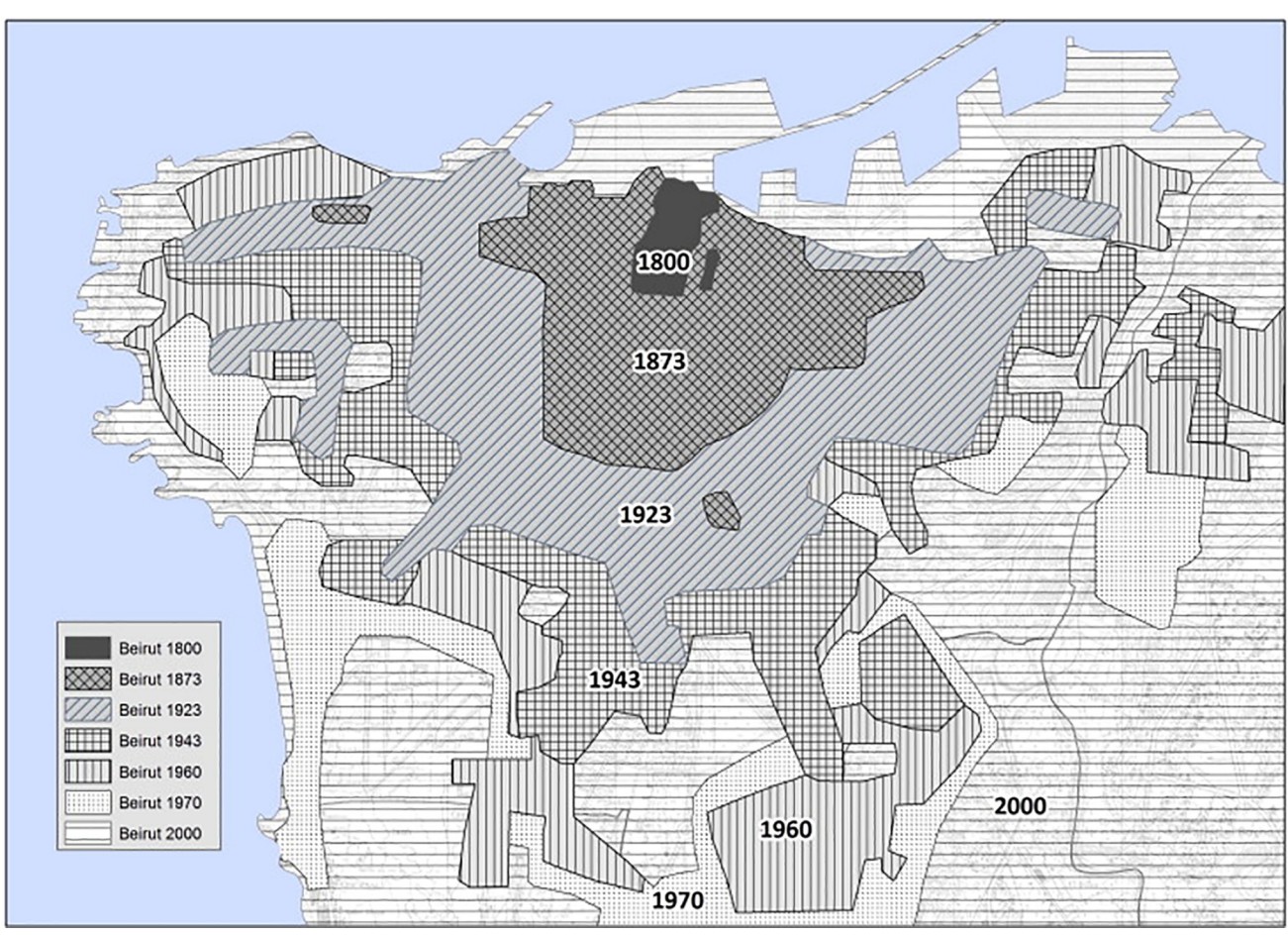

**Fig 1. Beirut phases of urbanisation (From "Beirut," by N. Yassin, 2010, Cities, 29, p. 64–73. Reprinted with permission.).**

intensity) of anthropogenic sites, and to levels of diversity of semi-natural habitats. Fig 2 presents the distribution of *M. crassifolia* in Beirut as well as the location of the selected sites for the study. All visited sites were either accessible public spaces or abandoned private lots. No permits were needed as no plant material was collected from study sites: only plant voucher specimens were collected from the field for taxonomic identification.

As our objective is to capture habitat diversity of a rare species in small plots,we used a deliberate biased method to select study locations and to lay out sampling quadrats [71]. In their attempt to compare the effects of random to non-random sampling on patterns of species abundance, species richness and vegetation-environment relationships, Diekmann et al. (2007) concluded that random sampling resulted in a larger number of common species, and a smaller number of rare species when compared to non-random sampling. They also found that for small plots, the number of species in the non-randomly placed plots was higher than in the randomly placed plots, and that in random sampling, there was considerable redundancy [105].

We set a total of 78 quadrats in 12 sites. In vegetation patches with clearly visible boundaries, one to two 1 m × 1 m quadrats were placed [32]. We placed larger quadrats, 2 m × 2 m, in locations where shrubs are present [32]. As in Dinsdale, deliberate bias method consists of placing quadrats in areas judged representative of the selected location and for capturing the maximum observed variation [71,106]. We made three modifications to the sampling

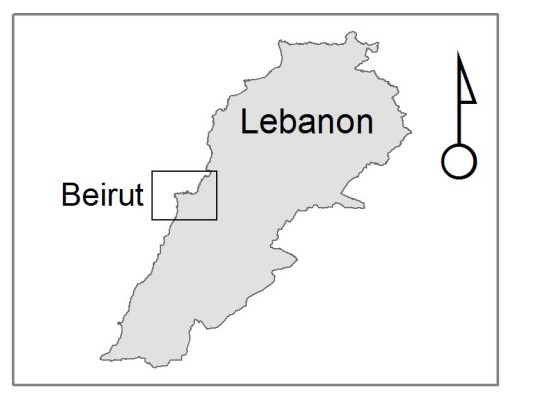

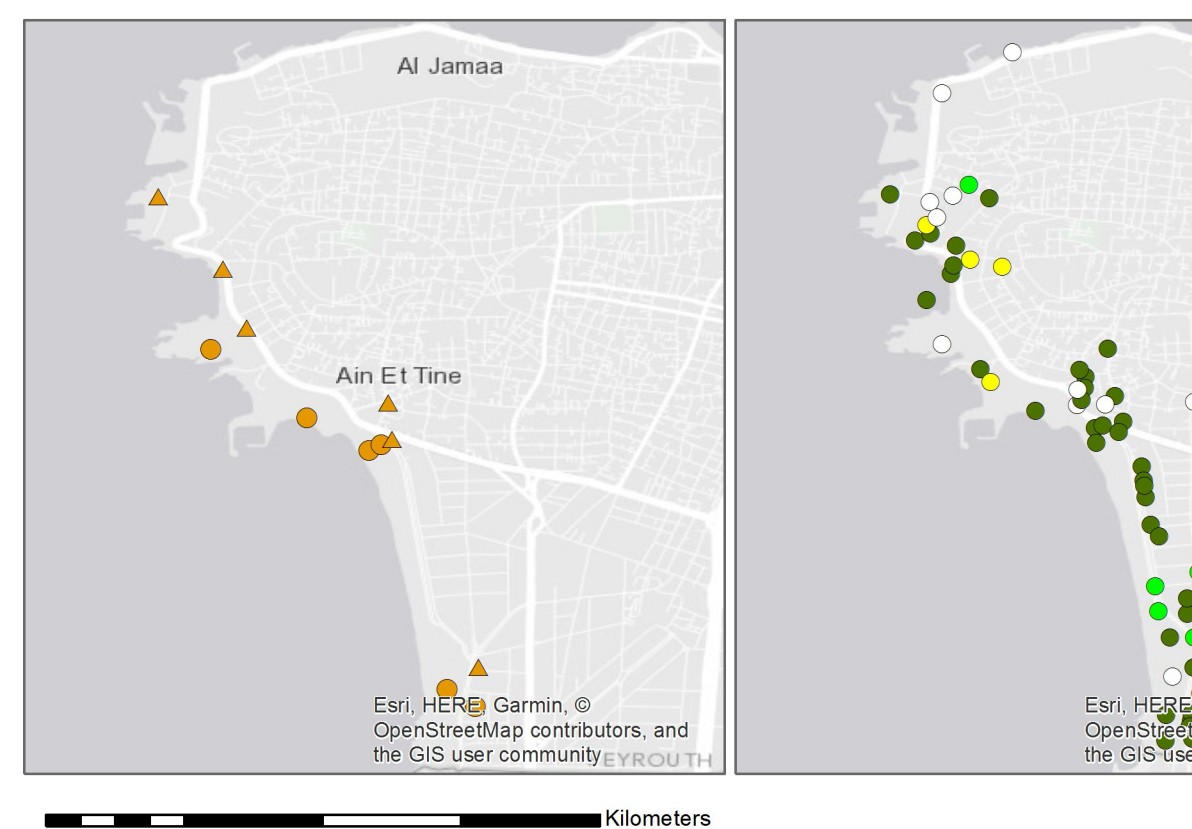

**Fig 2. The distribution of *M. crassifolia* in Beirut and selected sites for the study.**

technique to address site-specific challenges. First, when the boundary of a plant assemblage was not clearly defined due to site disturbance, we placed quadrats in locations where the target species was highly represented and at least one quadrat where the target species was minimally represented assuming that this location constitutes the boundary of the sampled plant assemblage. Second, when species had an 'individualistic' distribution pattern, we increased the number of quadrats, up to six, to capture the observed variation and compensate for the difficulty to define the boundaries of the plant assemblage [107]. Third, when the target species was consistently not present in a given vegetation assemblage, we placed one quadrat.

We divided each quadrat into a grid of 100 subunits to ensure speed of measurement and relative accuracy [108, 109, 110, 111]. In every quadrat, we determined percent cover using the

11-point Domin cover scale by visually assessing subunits as: fully covered, empty, and partially covered for each species and each life form [32]. Data obtained from all subunits within a quadrat was then added to determine Domin cover per quadrat.

As the analysis of non-randomly placed plots such as phytosociological quadrats may be biased, especially regarding estimates of species abundance and species richness patterns [105] ordination was not attempted to analyse vegetation-environment relationships in this study.

### Taxonomic and life form identification

We identified each plant specimen by consulting published floras, voucher specimens at the American University of Beirut Herbarium (Post Herbarium), and photographic floras [104, 98, 89]. All identified species were described by their life form according to Ellenberg and Mueller-Dombois, with amendment to include bunched shoot arrangement in reptant hemicryptophytes which forms a partially decomposed thick mat and causes peat accumulation [40]. We then pooled species that shared the same life form under the one category and estimated area cover for each life form accordingly.

### Analysis

Based on the 11-point Domin cover scale, we analyzed floristic data, species and percent species cover, using TWINSPAN [112]. Also called dichotomized ordination analysis, the Two Way INdicator SPecies ANalysis is a method for classifying communities according to hierarchical divisions based on progressive refinement of a single ordination axis of a (sites × species) data matrix [113]. Using the same tool, TWINSPAN, we analyzed the life form data, life-form categories and percent cover (as relative abundance of each life form within each quadrat). In the TWINSPAN, the cut levels 0-3-4-5-6-8 were applied. The TWINSPAN groups were characterized by constancy-percentage, average cover, and representation of target species. A matrix, integrating floristic and physiognomic TWINSPAN findings, was then created to find intersections between quadrat groups defined by classifying life form and floristic data sets. This process led to the identification of new classification that consisted of quadrat groups that share similar life form and species composition. The full dataset can be found in [92]. A conceptual extrapolation of these findings allowed us to define landscape plant typologies with vegetation assemblages similar to quadrat groups in which the target species is highly represented and we considered these typologies as suitable locations for the introduction of *M. crassifolia*.

## Results

*M. crassifolia* is most widely distributed in Beirut; based on our field surveys its presence was confirmed in 73 sites of which only one site, Pigeon Rock, is protected by law, and another site, the limestone cliff facing Pigeon Rock, is almost inaccessible and may be considered *de facto* protected. The remaining 71 sites offer highly diverse habitats and are not protected [92]. In remnant semi-natural sites, *M. crassifolia* is found in, spiny Mediterranean heaths, screes, sea cliffs and rocky offshore islands, growing on both sandstone and limestone formations and on (stabilized) coastal sand dunes. In anthropogenic sites, it grows near open sewers, in abandoned dump sites, through cracks in concrete walls and asphalt, on heaps of gravel, in street medians and on two occasions, almost epiphytically, out of the trunks of date and fan palms. The species' tendency to utilize modified habitats reflects its partial behavior as a ruderal [61]. During the course of this study, *M. crassifolia* was lost in 20 sites to urban development including one site which harbored the largest clump count, and only four of these sites were

recolonized. As a result, the plant species' range in Beirut was reduced by 17% between 2012 and 2015 [92].

We recorded the presence of 124 plant species belonging to 107 genera and 40 families in the 78 sampled quadrats [92]. Plant species co-occurring with *M. crassifolia* included 16% non-native species. Analysis of floristic data by TWINSPAN clustered the 78 quadrats under 17 quadrat groups labeled a[f] to q[f]. *M. crassifolia* had the highest constancy and abundance in three groups, d[f], g[f] and i[f]. In contrast, the species was not present in eight groups, c[f], f[f], k[f], m[f], n[f], o[f], p[f] and q[f]. The low community similarity, patchy species distribution, and predominance of habitat non-specific species reported by Talhouk et al. (2005) in their study of the floristics of the Lebanese coast was confirmed in this study [81]. High floristic variability between and within different sites resulted in more than half the quadrat groups (58.8%) consisting of no more than two quadrats. Only one group (e[f]) consisted of a large number of quadrats and represented a perceptible community of sparse vegetation on sandstone outcrops. Other groups were not site specific, but included quadrats exposed to similar disturbance; for example, in group g[f] the nine quadrats were sampled from street medians and side walks and consisted of a combination of evergreen exotic ornamental species such as *Agave americana*, *Agave attenuata*, and *Lampranthus multiradiatus*. Similarly, t[f] included quadrats characterized by a high representation of graminoids, *Cyperus rotundus* and *Cynodon dactylon*, which often grow in gardens and street medians under and around evergreen ornamentals such as the shrub *Pittosporum tobira*, and the creeping herbaceous forb *Sphagneticola trilobata*.

One problem we encountered with floristics based TWINSPAN analysis is that many groups did not represent actual communities i.e. plant species in an area that are unique and capable of coexisting as distinct, recognizable units that are repeated regularly in response to biotic and environmental variations [33, 34, 35, 36]. For example, group e[f], which included about 28% of sampled quadrats, consisted of several distinct vegetation assemblages that occur in different habitats, both semi-natural and anthropogenic, and the target species, a stress-tolerant ruderal, was the only common indicator species between these assemblages.

Life form description of plant species yielded 55 different life forms. Results revealed that more than half of all recorded species were therophytes with a total of 64 autotrophic therophyte and two heterotrophic annual vascular parasites. The high representation of therophytes reflects high disturbance of study sites [61].

Fig 3 presents the life-form spectrum of all species recorded in the 78 plots. Chamaephytes constituted the most prominent perennial life form and included 24 species. Over half of all chamaephytes were either regional or national endemics and only three were not native. Phanerophytes were represented by 14 species, 10 of which were not native. Perennials characterized by a periodic shoot reduction were represented by 15 hemicryptophytes and six geophytes.

Analysis of life form data by TWINSPAN clustered the 78 quadrats under 11 quadrat groups labeled

A[l] to K[l] (Table 1). *M. crassifolia* was highly represented in three of these groups (C[l], D[l], and E[l]) with a percent cover ranging between 11% and 25% in almost all quadrats within these groups. Examples of life forms in these three groups include, unbranched dwarf palm like trees (Phanerophyte08), typical and tall evergreen dwarf-shrubs (Chamaephyte03 & Chamaephyte04), low reptant evergreen succulents (Chamaephyte14), tall drought-deciduous hemicryptophytes (Hemicryptophyte01) and small reptant evergreen hemicryptophytes (Hemicryptophyte03) were common. Ornamental examples of these life forms include *Agave* and *Yucca* species (Phanerophyte08), cultivated Sea Lavender species (Chamaephyte03 and Chamaephyte04), and *Lampranthus multiradiatus* (Chamaephyte13).

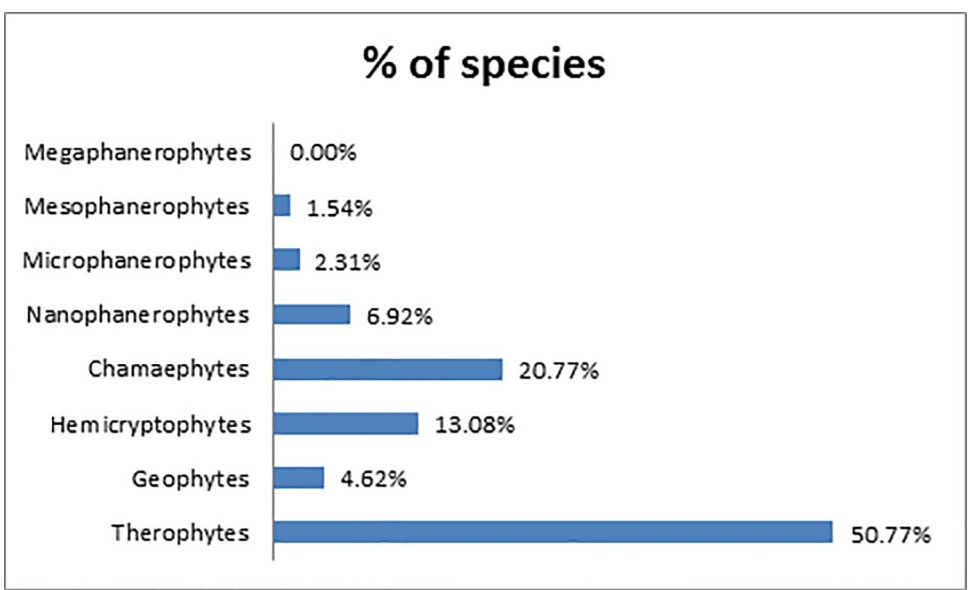

**Fig 3. Raunkiaer life-form spectrum of plant species recorded in 78 quadrats in 12 sites in Ras Beirut.**

Five groups (A[l], H[l], I[l], J[l], K[l]) excluded the target species and the dominant life form in these groups was mostly phanerophytes. These groups include mesophyllous large evergreen trees with spherical crown restricted to their upper half (Phanerophyte01), mesophyllous normal-sized evergreen shrubs with spherical crown extending to near their base (Phanerophyte04), microphyllous normal-sized evergreen shrubs with spherical crown extending to near their base (Phanerophyte03), and mesophyllous tall deciduous shrub with spherical crown extending to near the base of the shrub (Phanerophyte07). Ornamental examples of these life forms include various shade trees (Phanerophyte01), and shrubs used as hedges such as *Pittosporum tobira* (Phanerophyte04 and Phanerophyte03). They also include typical evergreen reptant herbaceous chamaephytes (Chamaephyte12) and ornamental plant species belonging to this life form and similar life forms such as turfgrass species and the Singapore Daisy, *Sphagneticola trilobata*.

The integration of floristic and life-form classification results into one matrix to identify quadrats at the intersection of both classifications generated a new set of quadrat groups that shared similar life form and species composition, and where *M. crassifolia* presented similar constancy and abundance (Table 2). This stepwise approach generated 30 quadrat groups, 8 which were highly favorable to *M. crassifolia*, and 12 which excluded it. We then proceeded to describe life form and species prevalent in these groups.

The intersections that resulted in quadrat groups with the highest representation of *M. crassifolia* belonged to 4 out of 11 quadrat groups that were derived from the classification of the life form data set (C[l], D[l], E[l] and F[l]) and 4 out of 17 quadrat groups that were derived from the classification of the floristic data set (a[f], d[f], g[f] and i[f]) (Table 3).

The intersections that resulted in quadrat groups with the lowest representation of the target species belonged to 8 out of 11 quadrat groups that were derived from the classification of the life form data set and 11 out of 17 quadrat groups that were derived from the classification of the floristic data set (Table 4).

## Discussion

The similarity in the infrastructure of a city may explain homogeneity of urban ruderal species, which out-compete sown species [114]. For example, a 30-year green roof study concluded

**Table 1. TWINSPAN analysis of life form data set collected in Ras Beirut.** (Alphabetical naming of quadrat groups by floristic and life form classification are not related.).

| | Quadrat groups (A to K) resulting from life form classification (l) | | | | | | | | | | |
|---|---|---|---|---|---|---|---|---|---|---|---|
| | A[l] | B[l] | C[l] | D[l] | E[l] | F[l] | G[l] | H[l] | I[l] | J[l] | K[l] |
| Phanerophyte08 | _ | _ | IV 5 | _ | _ | _ | _ | _ | _ | _ | _ |
| Phanerophyte09 | _ | V 5 | II 1 | _ | _ | _ | _ | _ | _ | _ | _ |
| Phanerophyte10 | _ | III 5 | II 3 | _ | _ | _ | _ | _ | _ | _ | _ |
| Chamaephyte13 | _ | III 6 | IV 5 | _ | _ | _ | _ | _ | _ | _ | _ |
| Hemicryptophyte12 | _ | III 1 | II 1 | _ | II 1 | _ | _ | _ | _ | _ | _ |
| Hemicryptophyte05 | _ | _ | II 1 | _ | I 2 | _ | _ | _ | _ | _ | _ |
| Chamaephyte05 | _ | _ | _ | II 6 | _ | _ | _ | _ | _ | _ | _ |
| Chamaephyte14 | _ | _ | _ | II 6 | _ | _ | _ | _ | _ | _ | _ |
| Hemicryptophyte03 | _ | _ | _ | III 4 | II 3 | _ | _ | _ | _ | _ | _ |
| Therophyte03 | _ | _ | _ | IV 2 | I 1 | _ | _ | _ | _ | _ | _ |
| Therophyte08 | _ | _ | II 1 | V 3 | I 1 | _ | _ | _ | _ | _ | _ |
| Therophyte02 | _ | _ | _ | IV 2 | III 1 | _ | _ | _ | _ | _ | _ |
| Phanerophyte07 | _ | _ | _ | _ | I 6 | _ | _ | _ | _ | _ | _ |
| Phanerophyte11 | _ | _ | _ | _ | I 3 | _ | _ | _ | _ | _ | _ |
| Phanerophyte12 | _ | _ | _ | _ | II 6 | _ | _ | _ | _ | _ | _ |
| Chamaephyte02 | _ | _ | _ | _ | I 5 | _ | _ | _ | _ | _ | _ |
| Chamaephyte09 | _ | _ | _ | _ | I 1 | _ | _ | _ | _ | _ | _ |
| Hemicryptophyte02 | _ | _ | _ | _ | II 2 | _ | _ | _ | _ | _ | _ |
| Hemicryptophyte06 | _ | _ | _ | _ | I 6 | _ | _ | _ | _ | _ | _ |
| Hemicryptophyte08 | _ | _ | _ | _ | III 3 | II 1 | _ | _ | _ | _ | _ |
| Hemicryptophyte09 | _ | _ | _ | _ | III 2 | _ | _ | _ | _ | _ | _ |
| Hemicryptophyte10 | _ | _ | _ | _ | I 2 | _ | _ | _ | _ | _ | _ |
| Geophyte01 | _ | _ | _ | _ | I 2 | _ | _ | _ | _ | _ | _ |
| Therophyte06 | _ | _ | _ | III 1 | IV 3 | II 1 | _ | III 2 | _ | _ | _ |
| Chamaephyte06 | _ | _ | _ | III 4 | II 4 | _ | III 2 | _ | _ | _ | _ |
| Therophyte01 | VI 1 | _ | _ | II 2 | I 2 | _ | _ | _ | _ | _ | _ |
| Therophyte04 | _ | _ | III 1 | VI 3 | IV 2 | III 1 | _ | _ | VI 2 | _ | _ |
| Hemicryptophyte01 | _ | _ | II 1 | II 2 | II 2 | _ | III 2 | _ | _ | _ | _ |
| Geophyte04 | _ | _ | III 3 | III 1 | | _ | _ | _ | IV 3 | _ | _ |
| Therophyte11 | _ | V 2 | V 2 | V 2 | IV 3 | IV 1 | _ | _ | _ | _ | _ |
| **Chamaephyte08** | _ | **VI 2** | **VI 4** | **V 4** | **VI 4** | **V2** | **IV 3** | _ | _ | _ | _ |
| Hemicryptophyte11 | _ | _ | _ | II 1 | IV 2 | III 2 | _ | _ | _ | _ | _ |
| Therophyte10 | _ | _ | _ | VI 2 | IV 2 | IV 1 | _ | _ | _ | _ | _ |
| Phanerophyte04 | _ | _ | _ | II 6 | I 3 | _ | _ | _ | IV 6 | _ | _ |
| Phanerophyte05 | _ | _ | _ | _ | II 5 | II 5 | _ | _ | _ | _ | _ |
| Chamaephyte01 | _ | _ | _ | III 4 | II 4 | II 3 | IV 4 | _ | _ | _ | _ |
| Chamaephyte04 | _ | V 4 | _ | II 4 | III 3 | III 3 | VI 6 | _ | _ | _ | _ |
| Chamaephyte07 | _ | _ | _ | _ | I 2 | II 1 | _ | _ | _ | _ | _ |
| Geophyte02 | _ | _ | _ | _ | III 2 | II 1 | _ | III 6 | _ | _ | _ |
| Therophyte05 | _ | _ | _ | IV 1 | IV 3 | V 3 | III 3 | VI 4 | _ | _ | _ |
| Therophyte09 | _ | _ | _ | II 2 | I 1 | _ | III 3 | _ | _ | _ | _ |
| Chamaephyte03 | _ | _ | _ | _ | I 3 | VI 4 | III 2 | III 3 | _ | _ | _ |
| Chamaephte12 | _ | _ | _ | _ | I 2 | II 1 | _ | _ | IV 6 | _ | _ |
| Geophyte03 | _ | _ | _ | _ | _ | II 2 | _ | _ | _ | _ | _ |
| Phanerophyte03 | _ | _ | _ | _ | _ | _ | _ | _ | _ | IV 6 | _ |

(*Continued*)

**Table 1.** (Continued)

| | Quadrat groups (A to K) resulting from life form classification (l) | | | | | | | | | | |
|---|---|---|---|---|---|---|---|---|---|---|---|
| | A[l] | B[l] | C[l] | D[l] | E[l] | F[l] | G[l] | H[l] | I[l] | J[l] | K[l] |
| Phanerophyte06 | – | – | – | – | – | II 2 | – | – | – | IV 6 | – |
| Chamaephyte11 | – | – | – | – | I 1 | II 1 | III 2 | VI 6 | VI 3 | VI 3 | – |
| Hemicyptophyte04 | VI 6 | – | – | – | – | – | – | – | – | – | – |
| Chamaephyte10 | – | – | – | – | – | – | – | III 4 | – | – | VI 2 |
| Phanerophyte01 | – | – | – | – | – | – | – | – | – | – | VI 6 |

The Roman number corresponds to species constancy within each TWINSPAN group (I = 5% or less; II = 6–20%; III = 21–40%; IV = 41–60%; V = 61–80%; VI = 81–100%). The Arabic number indicates average species abundance for each group on the domin scale (1 = less than 1%; 2 = 1–4%; 3 = 5–10%; 4 = 11–25%; 5 = 26–50%; 6 = 51–100%). Life-form of *M. crassifolia* is presented in bold and it is the only species under Chamaephyte08.

that spontaneous colonization should be accepted and considered as a design factor; and regional plant communities could serve as a model for seed recruitment and installations [114]. However, preventing a rapid loss of area-sensitive species necessitates large sites greater

**Table 2. Matrix of floristic and life-form classifications of quadrats from plant data set collected in Ras Beirut and southern part of the promontory of Beirut.**
Intersections show *M. crassifolia* represented by constancy and abundance and help define favorable and unfavorable vegetation assemblages.

| Quadrat groups (A[l] to K[l]) resulting from life form classification; Quadrat groups (a[f] to g[f]) resulting from floristic classification | A[l] | B[l] | C[l] | D[l] | E[l] | F[l] | G[l] | H[l] | I[l] | J[l] | K[l] |
|---|---|---|---|---|---|---|---|---|---|---|---|
| a[f] | – | – | – | 0 | VI 4 | – | 0 | – | – | – | – |
| b[f] | – | – | – | – | VI 3 | – | – | – | – | – | – |
| c[f] | – | – | – | – | 0 | – | – | – | – | – | – |
| d[f] | – | – | – | – | VI 4 | VI 4 | – | – | – | – | – |
| e[f] | – | VI 2 | – | – | IV 3 | V 2 | – | – | – | – | – |
| f[f] | – | – | – | – | – | – | – | – | 0 | – | – |
| g[f] | – | VI 3 | VI 4 | – | – | – | – | – | – | – | – |
| h[f] | – | – | – | – | – | – | VI 2 | – | – | – | – |
| i[f] | – | – | VI 5 | VI 4 | VI 4 | VI 4 | VI 3 | – | – | – | – |
| j[f] | 0 | – | – | VI 5 | VI 3 | – | – | – | – | – | – |
| k[f] | – | – | – | – | 0 | – | – | – | – | – | – |
| l[f] | – | – | – | VI 3 | – | – | 0 | – | – | – | – |
| m[f] | – | – | – | – | – | – | – | – | – | 0 | – |
| n[f] | – | – | – | – | – | – | – | – | 0 | – | – |
| o[f] | – | – | – | – | – | – | – | – | – | 0 | – |
| p[f] | – | – | – | – | – | – | – | – | 0 | – | – |
| q[f] | – | – | – | – | – | – | – | – | – | – | 0 |

Quadrat groups: A[l] to J[l] and a[f] to Q[f], f = floristic, l = life form (Alphabetical naming of quadrat groups by floristic and life fom classification are not related), constancy (I = 5% or less; II = 6–20%; III = 21–40%; IV = 41–60%; V = 61–80%; VI = 81–100%), average cover (1 = less than 1%; 2 = 1–4%; 3 = 5–10%; 4 = 11–25%; 5 = 26–50%; 6 = 51–100%).

**Table 3.** Description of urban plant habitat analogues (habitat condition, life forms, plant habitat, and species) for *M. crassifolia* in Beirut following a stepwise approach that intersects floristic and life form data classifications.

| Floristic classification | Life form classification | Average constancy and cover of target species | Habitat conditions | Description of urban habitat analogue: life form | Description of urban habitat analogue: Plant habitat and species |
|---|---|---|---|---|---|
| i[f] | C[l] | VI 5 | Semi-natural vegetation, mostly occupying coastal cliffs | Quadrat groups dominated solely by suffruticose chamaephytes, the life form of the target species, at an average cover of 11–50%, sometimes including fruticose chamaephytes or caespitose nanophanerophytes with scale like leaves, both at average cover of 26–50%. | The highest representation of the target species was only revealed through the matrix. The quadrat group shows that the target species probably prefers to be alone. |
| a[f] | E[l] | VI 4 | | | Species poor quadrat group. *M. crassifolia* was the only species consistently common between the quadrats. Perennials that less significantly occurred included *Thymbra capitata* and *Thymelaea hirsuta*. |
| g[f] | C[l] | VI 4 | Quadrat groups describing managed artificial vegetation of street medians | Low lying spreading succulent chamaephytes, at average cover of 26–50%, growing spontaneously or used as ground cover, sometimes interspersed by rosulate nanopherophytes, at average cover of 26–50%. Semi-rosette therophytes, at an everage cover of 1–4%, behaved as consistent ruderals. | Dominated by palm-like species of Agave and Yucca. *Lampranthus multiradiatus* used as ground cover. Several annuals, most notably *Urospermum picroides*, and *Matthiola crassifolia* behaved as ruderals. |
| i[f] | D[l] | VI 4 | | | *Polycarpon tetraphyllum* and *Crepis aculeata* were common ruderals—besides *Matthiola crassifolia*. *Carpobrotus edulis* dominated—*Pittosporum tobira* dominated once, but in that case, its canopy was disturbed. |
| d[f] | E[l] | VI 4 | Semi-natural vegetation, mostly occupying spontaneous urban wastelands | Very tall drought-deciduous scapose hemicryptophytes, at an average cover of 5–10%, as well as small and very tall scapose therophytes, at a cover that did not exceed 14%, were consistent ephemeral elements of this quadrat group. Shrubs such as tall evergreen semi-woody dwarf-shrubs and low (3–10 cm) creeping deciduous semi-woody dwarf-shrubs creeping along the ground were sometimes present at an average cover of 11–25%. Nanophyllous (usually less than 1 cm2) normal-sized evergreen shrubs sometimes dominated at an average cover exceeding 51%. | Sandy soil with small rock fragments sometimes alternatively dominated by *Dittrichia viscosa*, *Thymaleae hirsuta* or *Convulvulus secundus*, among other perennials and annuals, but consistently including the target species as well as *Alcea setosa* |

(*Continued*)

**Table 3.** (Continued)

| Floristic classification | Life form classification | Average constancy and cover of target species | Habitat conditions | Description of urban habitat analogue: life form | Description of urban habitat analogue: Plant habitat and species |
|---|---|---|---|---|---|
| i[f] | E[l] | VI 4 | Quadrat groups of samples collected from minimally managed artificial vegetation of street medians and highly disturbed semi-natural patches | Drought deciduous semi-rosette scapose hemicryptophytes at an average cover of 5–10% and tall scapose therophytes at an average cover of 5–10% were regular features in this group of quadrats. Besides graminoid phanerophytes being seldom present at an average cover exceeding 50% and thus behaving as dominant evergreen perennial elements, suffrutescent chamaephytes were consistently present at an average that did not exceed 25%. | This quadrat group included both anthropogenic and disturbed semi-natural habitats. Sparse vegetation composed of evergreen ornamentals and ruderals growing on a mostly bare sandy soil mixed with gravel in a minimally managed street median or cracks in concrete. Vegetation growing on slightly stabilized sands of a sandy beach; meeting line of sandstone formation with pedestrian path, abandoned dump site; mostly bare ground on wet sandstone cliff occupied by sparse vegetation; mostly bare ground on wet sandstone cliff occupied by sparse vegetation; part of steep sandstone cliff dominated by *Galium canum*; sandy soil with small rock fragments and cement dominated with *Arundo donax* |
| i[f] | F[l] | VI 4 | Abandoned anthropogenic structures | Typical or tall caespitose and tall scapose suffrutescent chamaephytes codominating vegetation at an average abundance of 26–50%. | Crack in concrete through which few perennial species grow; A bolder protruding from a sandstone cliff allowing for both *Limonium mouterdei* and *Matthiola crassifolia* to grow on it; Part of steep sandstone cliff dominated by *Galium canum* |
| d[f] | F[l] | VI 4 | Abandoned part of public beach | | *Dittrichia viscosa* and *Matthiola crassifolia* dominating vegetation growing on slightly stabilized sands of a sandy beach |

Alphabetical naming of quadrat groups by floristic and life form classification are not related.

than 50 ha [88]. Such areas are absent in Beirut, and the loss of *Matthiola crassifolia* in the city is highly likely. Utilizing habitat analogues to increase the area of habitat patches and create a network of corridors is the most plausible strategy to ensure the persistence of this narrow endemic.

Urban environments share many features in common because they are designed to perform standard functions to meet human needs [115]. Such an environment, common to cities around the world, might be expected to select for species with a similar suite of traits favouring persistence in highly disturbed and human-modified habitats. Indeed the process of urbanization has been conceptualized as a series of filters acting on an existing species pool and selectively removing those species with traits unfavourable for persistence in this new environment [59].

The peculiarity of our study is that, not only is classification influenced by ruderals, but the species of conservation interest *M. crassifolia* also behaves as a ruderal. Considering the diversity of habitats occupied by *M. crassifolia*, it was not possible to resolve this lack of location specificity with floristic assessments, which in turn did not allow us to develop an understanding of urban habitat analogues. Instead, the number of quadrat groups generated by the

**Table 4. Description of urban plant habitats (habitat condition, life forms, plant habitat, and species) unsuitable for *M. crassifolia* in Beirut following a stepwise approach that intersects floristic and life form data classifications.**

| Quadrat Group by Floristic Classification | Quadrat Group by Life form Classification | Description of the intersecting groups that exclude *M. crassifolia* | Description of habitats and species of the intersecting groups that exclude *M. crassifolia* |
|---|---|---|---|
| a[f] | G[l] | Natural assemblages dominated by suffruticose chamaephytes at an average cover exceeding 51%, sometimes also dominated by fruticose chamaephytes | *Galium canum* growing as clumps on steep sandstone cliff |
| l[f] | G[l] | | *Crithmum maritimum* growing on slightly stabilized sand beach |
| a[f] | D[l] | | *Thymbra capitata* dominating a limestone formation |
| j[f] | A[l] | Natural and artificial assemblages dominated by thick mat-forming reptant herbaceous hemicryptophytes or chamaephytes at an average cover exceeding 91%; sometimes geophytes were significantly present | *Phyla nodiflora growing as thick mat* |
| c[f] | E[l] | | Sandy soil ground covered with some sandstone pebels and a thick layer of reptant herbaceous plants such as *Polygonum equisetiforme* among which many annuals. |
| f[f] | I[l] | | Street median dominated by *Sphagneticola trilobata* |
| n[f] | H[l] | | Sandy soil and degraded limestone or sandstone dominated by dense creeping Sporobolus pungens and *Cynodon dactylon*, sometimes high *Oxalis pes-caprae* |
| p[f] | I[l] | Artificial and spontaneous vegetation assemblages dominated with microphyllous and mesophyllous mostly evergreen normal-sized and tall shrubs as well as large sized trees at an average cover exceeding 91%. | Hedge of *Pittosporum tobira* in garden of a residential building |
| m[f] | J[l] | | *Lantana camara* in residential gardens |
| o[f] | J[l] | | Street median entirely covered with *Carissa macrocarpa* |
| k[f] | E[l] | | *Paritaria judaeca* and *Ricinus communis* growing as understory of *Ficus carica* along an open sewer |
| q[f] | K[l] | | Tufts of *Piptatherum miliaceum* growing on sandy soil and rubble under a canopy of *Ficus microcarpa* |

Alphabetical naming of quadrat groups by floristic and life form classification are not related.

floristic analysis was large, and some of these clusters did not represent actual plant community assemblages. Although the natural habitat of the target species is described as coastal area rocks [89], the behavior of the target species as a ruderal led to TWISPAN quadrat groups of highly variable species constituency and quadrat locations ranging from highly managed street medians to semi-natural coastal cliffs and including commercially introduced species and native ones.

Classifying life form data by including percent cover for each category helped specify which life forms and their respective abundance were positively or negatively associated with *M. crassifolia*. Our findings are in line with Kent [32], who emphasized that physiognomy might be more useful as a tool than floristics in highly modified habitats at different scales due to the responses of plant species to macro- and micro-climate conditions. Life history and life form are stronger predictors of underlying population processes than native status [87,116].

By using a stepwise approach which combines the two methods, floristics and physiognomy, we were able to minimize the masking effect of ruderal species and to identify life form similarities within distinct vegetation assemblages. In the last decade, researchers have combined life form and floristic vegetation description methods to overcome difficulties in analyzing data in disturbed habitats. For example, Vestergaard [117] generated quadrat groups based on floristic data through TWINSPAN and then described the life-form spectra in each to investigate the relationship between plant diversity and artificial dune development processes. Although similar to our methodology, Vestergaard did not use this combined methodology to define habitat analogues for target plant species. In 2014, a new vegetation classification approach that relies on both physiognomy and floristics over large areas was published under

the name EcoVeg [48]. Our approach, however, differs from EcoVeg in that we first mathematically classify physiognomic data and later sort the classifications according to a specific floristic trend. In addition, we base our study on field data collected from small urban habitat sites while EcoVeg uses map data and is meant to classify vegetation over large natural landscapes.

More recently, several studies have sought to explore the potential of light detection and ranging (LiDAR) to inform landscape biodiversity assessments. In fact, the utilization of this technology has developed from quantification of gaps (above bare ground, low vegetation and medium vegetation), canopy cover and its vertical density in open landscapes [118] to mapping tree cover and vegetation spatial and vertical structure in cities and estimating above ground biomass despite particular challenges posed by urban areas [119,120].

Furthermore, accurate mapping of vegetation communities within highly disturbed urban landscapes was recently achieved through incorporating a hierarchical object-based image analysis (OBIA) approach with high-spatial resolution imagery and canopy height surfaces derived from LiDAR data [121]. Provided the range of outputs these recent methods are producing, LiDAR technology may serve for rapid indentification of potential locations for habitat analogues and the exclusion of areas that are known not to be favorable to the target species, for example canopy cover in the case of *M. crassifolia*.

Improving the quality of existing green spaces throughout the entire urban matrix has been suggested as an effective approach to enhancing biodiversity experience [122]. The information we generated using a stepwise approach integrating floristics and physiognomy, may serve as blueprints for planting designs; it offers a plant selection palette that is not restrictive and does not enforce a native only policy. The habitat conditions in quadrat groups of high representation of the target species were not the same and reflected a wide range of potential habitat analogues for *M. crassifolia*. These varied from abandoned buildings to highly managed street medians.The urban habitat analogues that we identified include green spaces dominated by palms, low-lying succulents, or shrubs with scale-like leaves. In contrast, the species does not seem to persist in green spaces dominated by turf grass, canopy trees, or vegetation that produces a significant litter. Furthermore, since knowledge of a target species' preferred physiognomies includes an understanding of its position in the vertical stratification of its ecological community [32], we were able to identify additional habitats suitable for the introduction of *M. crassifolia*. Streetscapes, such as street medians, sidewalks and street tree planters, that lack both peat accumulating ground cover and canopy species, ubiquitous throughout the city, provide patches of optimal vegetation composition that could potentially accommodate *M. crassifolia*. Such streetscapes that can function as habitat analogues for *M. crassifolia* are illustrated in Fig 4 ana Fig 5. A change in landscape management strategy, however, needs to preceed design and development of habitat analogues. At the end of the four-year study, *M. crassifolia* was no longer seen in 16 out of 73 sites. Our field observations, revealed that management strategies such as the intentional uprooting of M. crassifolia considered by gardeners as a weed led to the disappearance of the species from these locations. On the other hand, there are locations where the species persists probably due to the fact that in these sites gardeners remove plants during their dieback stage, which includes seed-bearing silique fruits, but they keep seedlings and flowering plants.

In the case where more than one species is a conservation target, then a conservation strategy conducive to the persistence of both species. In our study, we found that *M. crassifolia* persisted as part of the low shrub layer under taller nanophyllous shrubs like the Shaggy sparrowwort, *Thymalea hirsuta*, another species of conservation interest in Lebanon. *M. crassifolia* also thrived in the understory of tuft-trees like the fan palm, *Washingtonia robusta*, an introduced species, and within groves of the giant reed, *Arundo donax*, a spreading native. Species

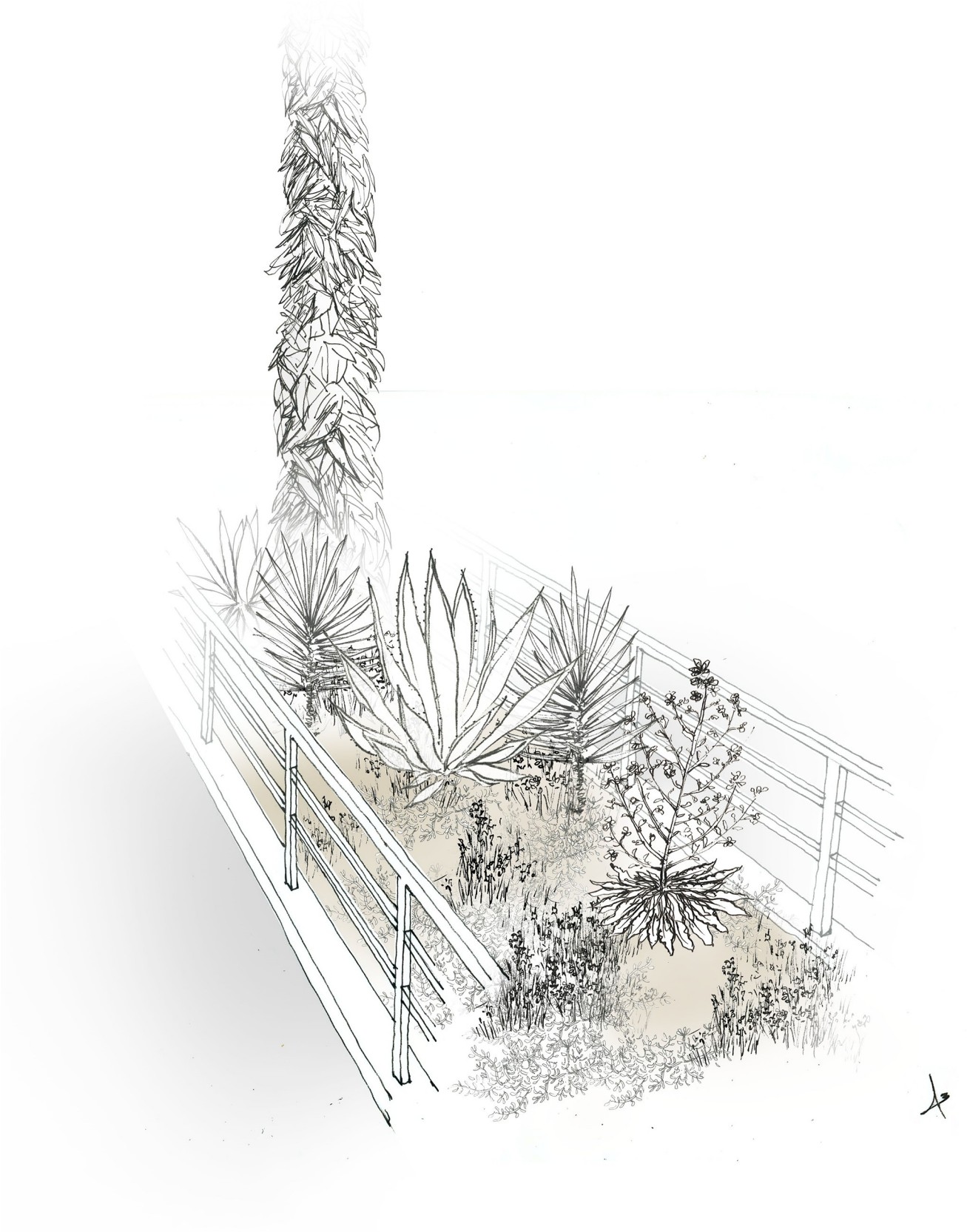

**Fig 4. Illustrated scene for a planted street median functioning as a habitat analogue for *M. crassifolia*.** Rosulate phanerophytes and reptant succulent chamaephytes, often used as ornamentals in green spaces in Beirut, dominate the street median without excluding the target species.

belonging to these life forms, or similar ones, dominate many sites in Beirut including street medians and could serve as favorable habitats for *M. crassifolia*. Our findings also show that some exotic invasive species impacted *M. crassifolia* positively. *M. crassifolia* grew in sites dominated by *Carpobrotus edulis*, a potentially invasive in Lebanon, planted at the edge of pedestrian paths. Pedestrians avoided stepping onto these areas, maybe due to their appreciation of *C. edulis* as an evergreen ground cover [123]. As a result, this plant assemblage protected *M. crassifolia* and allowed *C. edulis* to spread constrained by water availability. Removal of invasive plant species should be determined based on its impact on endemic and rare vegetation present in a given region, and eradication should focus on those invasive species that compete with endemic species in general and those of conservation interest especially [124]. Huenneke and Thomson [125] suggest criteria for determining whether such species pose problems for specific rare native taxa and indicated the possibility that some species may be beneficial to endemics.

Equipped with the findings above, landscape designers, architects, and managers can better reconcile between desired conservation targets and, socio-behavioral, and aesthetic outcomes by including *M. crassifolia* in an aesthetically pleasing setting. They can design urban habitat analogues that promote the persistence of *M. crassifolia* by excluding from the plant palette native or non-native species belonging to life forms associated with its low representation as

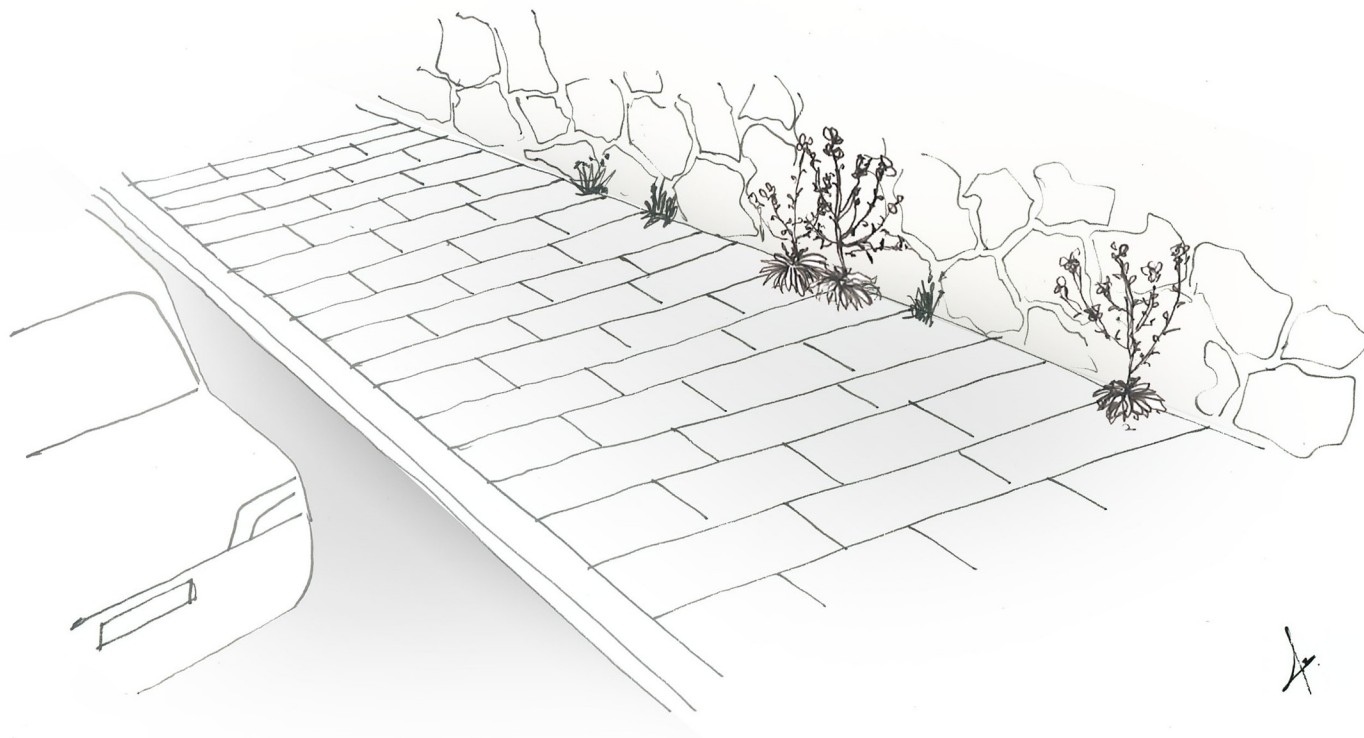

**Fig 5. Illlustrated scene for sidewalk functioning as a habitat analogue for *M. crassifolia*.** The cracks in the concrete of the sidewalk due to poor management and the adjacent sandstone wall resemble coastal cliffs occupied by the species. Small and medium-sized therophytes like *Plantago coronopus* L. abd *Polycarpon tetraphyllum* are often observed occupying such spaces.

reported in this study. Alternatively, they can design an urban habitat analogue using a vegetation architecture conducive to the persistence of *M. crassifolia*. In the case of established green spaces, they can manage the space to become suitable for *M. crassifolia* by selectively removing species with a life form that is incompatible or that restricts its abundance. In some situations, horticultural techniques, such as pruning, can modify the micro environment without changing species existing on site, to create suitable urban habitat analogues; for example, improving light conditions in cases where species of conservation interest is shade intolerant.

Identifying predictable relationships between plant traits and environmental conditions provides a promising framework for understanding how vegetation responds to environmental change in a variety of ecosystems [21].

## Conclusion

Given the rate of expansion of urban landscapes [126, 127, 128, 129], increasing a target species' site area in a city is highly desired [28]. Our findings may serve as guidance on how to create or modify, through landscape planting designs, suitable habitats for species of conservation interest. By understanding the physiognomy and structure, and environmental conditions in which a species occurs, green areas may be designed to suit the requirements of a target species while established areas may be surveyed for candidate sites suited for the introduction of a target species. Our stepwise approach offers a detailed field assessment tool for urban plant habitat analogue characterization.

## Supporting information

**S1 Data.**
(XLSX)

**S2 Data.**
(TXT)

**S3 Data.**
(TXT)

**S1 Material.**
(DOCX)

**S2 Material.**
(DOCX)

**S1 File.**
(PDF)

## Acknowledgments

This paper is derived from the dissertation submitted by M. Itani in partial fulfillment of the requirements
for the MSc. degree at the American University of Beirut. We thank Drs. R. Zurayk, N. Farajalla, and K. Knio for their inputs throughout the study. We thank K. Mohamed, A. Jammool, S. El Masri, O. El Tal, R. Atallah and N. Halabi for their in field data collection. We thank M. Bou Kanaan for producing the illustrations.

## Author Contributions

**Conceptualization:** M. Itani, M. Al Zein, S. N. Talhouk.

**Data curation:** M. Itani.

**Formal analysis:** M. Itani.

**Investigation:** M. Itani.

**Methodology:** M. Itani, S. N. Talhouk.

**Project administration:** S. N. Talhouk.

**Resources:** S. N. Talhouk.

**Software:** M. Itani.

**Supervision:** S. N. Talhouk.

**Visualization:** M. Itani, N. Nasralla, S. N. Talhouk.

**Writing – original draft:** M. Itani, N. Nasralla, S. N. Talhouk.

**Writing – review & editing:** M. Itani, N. Nasralla, S. N. Talhouk.

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
