## [Decision Letter · Decision Letter 0]

17 Jan 2020

PONE-D-19-19705

Biodiversity conservation in cities: Defining habitat analogs for plant species of conservation interest

PLOS ONE

Dear Professor Talhouk,

Thank you for submitting your manuscript to PLOS ONE. After careful consideration, we feel that it has merit but does not fully meet PLOS ONE’s publication criteria as it currently stands. Therefore, we invite you to submit a revised version of the manuscript that addresses the points raised during the review process.

Most importantly, the study should be better linked to recent concepts and results of urban ecology, thereby making use of the broad literature in this field of research. For instance, this includes current knowledge about conservation in urban areas, applying the concept of novel ecosystems, using plant functional traits, and referring to approaches for classification of habitat types in other cities. Many of your statements are not supported by literature, e.g. why vegetation physiognomy or structure should be more useful for urban biodiversity research than floristics.

One important methodological question is why you did not take into account local habitat conditions when assessing habitat analogues for a species of conservation concern; it would be important to include, for instance, abiotic site conditions or management intensity at these sites. Another methodological issue refers to the non-random location of plots, which will have major implications for the interpretation of data. Likewise, the target species’ habitat preference can hardly be assessed if it occurs on nearly all sites.

In general, some basic information is missing in the Materials and Methods chapter, including the criteria used for site selection and how you identified the additional habitats. The reviewers also provide a range of recommendations regarding the presentation of results. For instance, a more consistent coding of life-form types has to be used and definitions of life-form types has to be added; the number of tables should be reduced; some results may be shown using graphical presentations.

There are many other helpful comments, in particular provided by reviewers 2 and 3 that should be considered when revising the manuscript.

We would appreciate receiving your revised manuscript by Mar 02 2020 11:59PM. To enhance the reproducibility of your results, we recommend that if applicable you deposit your laboratory protocols in protocols.io, where a protocol can be assigned its own identifier (DOI) such that it can be cited independently in the future. For instructions see: http://journals.plos.org/plosone/s/submission-guidelines#loc-laboratory-protocols

We look forward to receiving your revised manuscript.

Kind regards,

Harald Auge

Academic Editor

PLOS ONE

Journal Requirements:

Additional Editor Comments (if provided):

Reviewers' comments:

Reviewer's Responses to Questions

**Comments to the Author**

1. Is the manuscript technically sound, and do the data support the conclusions?

Reviewer #1: Yes

Reviewer #2: Yes

Reviewer #3: Partly

2. Has the statistical analysis been performed appropriately and rigorously? 

Reviewer #1: Yes

Reviewer #2: Yes

Reviewer #3: N/A

3. Have the authors made all data underlying the findings in their manuscript fully available?

Reviewer #1: Yes

Reviewer #2: Yes

Reviewer #3: Yes

4. Is the manuscript presented in an intelligible fashion and written in standard English?

Reviewer #1: Yes

Reviewer #2: Yes

Reviewer #3: Yes

5. Review Comments to the Author

Reviewer #1: The manuscript includes a novel stepwise approach in identifying habitats suitable for target species. The use of a combination of lifeform spectrum snd physiognomic characteristic and floristic identity has helped in proposing management strategies for Matthiola crassifolia. I admire the approach

Reviewer #2: INTRODUCTION

1. The introduction is missing citations, as many statements are not supported by literature.

2. I would only use the world “artificial” (e.g. line 76) to describe urban habitats that have been deliberately designed by man with a specific purpose. Not all urban habitats should be classified as such.

3. Before introducing Table 1, the authors mention that current data collection methods are designed for non-urban contexts. However, they should be aware that very important work has been done to classify the urban habitats inside the city of Berlin. This project lasted several years and was done by the Senate Department for Urban Development and Housing of Berlin (more info here: http://www.stadtentwicklung.berlin.de/umwelt/umweltatlas/ed508_03.htm). They differentiated more than 7000 different urban biotopes/habitats inside the city of Berlin.

4. I would include in the introduction an hypothesis stating what kind of analogue habitats the authors expect to find for M. crassifolia.

5. In general, I have the feeling that this section could profit from a deeper insight into current findings in urban ecology. This is a non-exhaustive list of some important literature that could help:

• Anderson, EC & Minor, ES (2019) Assessing social and biophysical drivers of spontaneous plant diversity and structure in urban vacant lots. Sci Total Environ, 653, 1272-1281.

• Beninde, J, Veith, M & Hochkirch, A (2015) Biodiversity in cities needs space: a meta-analysis of factors determining intra-urban biodiversity variation. Ecol Lett, 18, 581-592.

• Chocholoušková, Z & Pyšek, P (2003) Changes in composition and structure of urban flora over 120 years: a case study of the city of Plzen. Flora, 198, 366-376.

• Duncan, RP et al. (2011) Plant traits and extinction in urban areas: a meta-analysis of 11 cities. Global Ecology and Biogeography, 20, 509-519.

• Ives, CD et al. (2016) Cities are hotspots for threatened species. Global Ecology and Biogeography, 25, 117- 126.

• Knapp, S et al. (2008a) Urbanization causes shifts of species' trait state frequencies. Presilia, 80, 375-388.

• Knapp, S et al. (2009) How species traits and affinity to urban land use control large-scale species frequency. Diversity and Distributions, 15, 533-546.

• Kowarik, I & von der Lippe, M (2018) Plant population success across urban ecosystems: A framework to inform biodiversity conservation in cities. Journal of Applied Ecology, 55, 2354-2361.

• Planchuelo, G, von der Lippe, M & Kowarik, I (2019) Untangling the role of urban ecosystems as habitats for endangered plant species. Landscape and Urban Planning, 189, 320-334.

• Schmidt, KJ, Poppendieck, HH & Jensen, K (2014) Effects of urban structure on plant species richness in a large European city. Urban Ecosystems, 17, 427-444.

• Shwartz, A et al. (2014) Outstanding challenges for urban conservation research and action. Global Environmental Change, 28, 39-49.

METHODS

6. From my perspective, the description of the genus and the whole range of species beyond M. crassifolia is not necessary.

7. In lines 131 to 135, I would state that the mentioned sites are cities. Many readers might not be familiar with the geography of Lebanon. Additionally, I would use the English instead of the local name of the city (i.e. Sidon instead of Saida) for international readers.

8. I don’t fully understand what the authors wanted to achieve with point 3 starting in line 170. Where was the second quadrat placed and why?

RESULTS

9. I would argue that this section in general needs to be easier to interpret and more visual. I think most tables don’t necessarily need to be included in the results. Perhaps I would include them as an annex and only represent a summary of some of them in the results. For example, Table 2 could be summarized showing only the genus that co-occur with M. crassifolia, not all the species, or Table 4 could be easily summarized with a pie chart showing the different proportion of each life-form. Additionally, some data mentioned in the text could be represented visually in a graph or similar (i.e. the proportion of M. crassifolia found in remnant sites vs. anthropogenic areas, or a map showing the location of these 20 sites where M. crassifolia was lost)

DISCUSSION

10. I think the authors should discuss about their own results earlier in this section. Perhaps some of the text between lines 353-396 could go to the introduction.

11. It could be interesting to include some discussion about the characteristics of the sites where M. crassifolia was lost.

12. I think it could be interesting to discuss about the ecological reasons explaining why there is a difference in the presence of M. crassifolia in remnant sites vs. anthropogenic sites.

13. Regarding the non-random design of this study, it might be interesting if the authors would consider this paper when interpreting their results: Diekmann, M., Kühne, A. & Isermann, M. Folia Geobot (2007) 42: 179. https://doi.org/10.1007/BF02893884. It shows that in small plot sizes like the ones used in this study, non-randomly placed plots result in general in higher species richness than if they were randomly distributed.

14. As a general comment for this section, I would argue that in order to develop a methodology to assess the habitat analogues of a species, the authors should consider including an assessment of not only floristics and physiognomy, but also of the local habitat conditions. As an example, we could reflect on the areas where M. crassifolia thrives (those with palms, succulents or dry shrubs) - perhaps there is something in these locations beyond the species assemblages that can explain why the target species is performing favorably. Are these locations the least managed? Are they in spontaneous urban wastelands? I think it is very important that the authors include an additional assessment of the local habitat conditions in their study.

GENERAL REMARKS

I think more work is needed to improve the scientific quality of this paper. The authors should make use of the broad literature available in the field of urban ecology and their methodology should include an assessment of the local habitat conditions when assessing habitat analogues for a species of conservation concern.

Reviewer #3: Dear authors of „Biodiversity conservation in cities: Defining habitat analogs for plant species of conservation interest“,

The idea of identifying habitat analogs for species of conservation interest within human-dominated landscapes is an interesting idea for improving species conservation and, in your case, for improving the contribution of urban areas to the protection of biological diversity.

After reviewing your manuscript, I have a number of concerns that in my opinion should be solved before the manuscript is ready for publication. I will describe my concerns below, point by point.

Abstract:

- Line 28: I am not sure what you want to express with the first sentence of the abstract. Do you want to express that urban environmental conditions change the composition of species assemblages or that species in urban areas interact differently with each other than they do within non-urban habitats? If you mean the latter, is there proof for it? For example, would two species interact differently with each other depending on whether they occur in an urban or in a non-urban area?

- You distinguished among native, naturalized, ornamental garden escapes and invasive plant species (e.g., lines 29/30, 73, 145). Could you state more precisely that these are native and non-native species, with the three latter groups comprising non-native species? Also, I’d rather talk about casual non-native species than about ornamental garden escapes because usually, non-naturalized non-native species in urban areas do not only (although to a large share) include ornamental garden escapes but also other introduced species that have not naturalized (yet), such as species accidentally introduced.

- Line 34: Where are these 12 study sites located? Please say so in the abstract and consider adding a figure to the manuscript that depicts the location of sites.

Introduction: The introduction would benefit from better linking it to recent urban biodiversity research, especially to our knowledge about species conservation in urban areas, to the concept of novel ecosystems, and to functional traits and functional groups:

- The persistence of species of conservation interest within urban areas is not in general unlikely (and thus I do not completely agree with your sentence in lines 75/76). A range of threatened species can occur within urban areas (see Ives et al. 2016, Global Ecology and Biogeography 25, 117-126). Whether a species will be able to persist within urban environmental conditions does depend on its functional traits (cf. Williams et al. 2015, Perspectives in Plant Ecology, Evolution and Systematics 17, 78–86).

- Introduction, lines 73/74: Well-established standard classifications of urban habitats exist, such as the four types of urban nature by Kowarik (1992 – cited e.g. in Kowarik 2018 Urban Forestry & Urban Greening 29, 336-347). You do not necessarily need to refer to one of these classifications, but please note that urban areas usually not only comprise artificial habitats but also remnants of natural habitats.

- Lines 84-86: I do not see why the methods used to describe vegetation that you present should only work in natural areas. I guess that what you want to express is that the TWINSPAN-classification of groups does not work that well because, as you wrote later, there are many unique co-occurrences of plant species at different urban sites. However, the methods you present in Table 1 can be applied in all kinds of terrestrial plant communities, including urban ones. It’s just that the results differ among urban and non-urban habitats. Concerning the many unique co-occurrences of plant species at different urban sites, I suggest that you refer to the concept of novel ecosystems (see Hobbs et al. 2006, Global Ecology and Biogeography 15, 1-7 for the overall concept and Kowarik 2011, Environmental Pollution 159, 1974-1983 for a review on novel urban ecosystems) because what you describe – untypical, unique species assemblages – might exactly be this – novel urban ecosystems.

- Table 1: How did you collect / choose these methods? Did you perform some kind of literature review to do so? If not, how do you know that the overview is complete? Also, I do not see much use in presenting all this detailed information. The table occupies a lot of space but the basic message could be wrapped into one or two sentences: The methods you present in Table 1 describe vegetation based on species identity and abundance, species functional traits, structural characteristics or degree of naturalness. All other details are not necessary for understanding your study and interested readers can refer to the literature cited. So, my suggestion is to delete Table 1 and to wrap its most important content into one or two sentences with references. (cf. Discussion, line 353 where you wrote about “two main vegetation description methods” – so, this can be nicely wrapped)

- Citations in lines 94/95 – a lot. Maybe it is not necessary to cite all of these 17 references. Rather choose some and provide them as example references.

- Lines 95/96: please, do provide references for the statement that physiognomic and structural vegetation description may be a more useful tool for urban biodiversity than floristics. To better link your manuscript to recent urban biodiversity research, I think it is useful to refer to functional plant traits. The physiognomic characteristics that you look at equal functional traits and functional groups. (See e.g. Williams et al. 2015, Perspectives in Plant Ecology, Evolution and Systematics 17, 78–86 and Lavorel et al. 1997, Trend in Ecology and Evolution 12, 474-478). Especially, to relate species to environmental conditions or their ability to co-occur with other species, trait-based approaches (or physiognomic approaches, as you call them) are indeed more helpful than floristic approaches.

- In addition, some information on the target species and why it is the target species of your study should be added to the Introduction. From all the data that you present, I got the impression that Matthiola crassifolia performs reasonably well in the urban area of Beirut. So, is it really threatened by urbanization? Also, in which natural habitats does the target species usually occur? Please, add this information to the description of the study species as it seems basic to understanding habitat analogs (as these will be analog to the natural habitats of the target species, won’t they?)

Materials and Methods:

- The identification of habitat analogs for the target species is solely based on the identification of species and life form types it occurs with. No abiotic conditions (temperature, soil type, soil moisture, pH, nutrients, surrounding land cover, …) have been taken into account although the occurrence and abundance of a species is not only determined by its biotic interactions but also, to a major extent, by abiotic conditions. Why haven’t abiotic conditions been taken into account?

- How were the 12 study sites selected? Was selection based on specific criteria? Which data (such as habitat or land-use maps) were used to select sites? I do understand the method that you used, still, it does not explain to me how sites were selected, i.e.: Were all anthropogenic and semi-natural habitats that occur in Ras Beirut taken into account or were specific anthropogenic and semi-natural habitats selected? This question is crucial because in the end, there are 78 plots and in 73 of these plots, the target species occurs. How can you, statistically sound, assess the preference of the target species for specific species assemblages if you have hardly any sites where it does not occur?

- Basic temporal information is missing in the Materials and Methods chapter: How long did the study last? When did the study start? When did the study end?

- Lines 122-124: Why are the species group names written in large letters?

- Line 127: add “should” among “species proposing that it” and “be considered”

- Line 133: “three out of five previously reported sites, Beirut and Byblos”. These are just two sites, not three. What’s the third one?

- Lines 134/135: What about Amchit? Has it been extirpated there as well or does it still occur there?

- Lines 142: With 20 km², you refer to the city, not to the total metropolis of Beirut, right? Please specify.

- Lines 156: Is there a reference for the early botanical studies of semi natural areas in Beirut? Also, when did these studies take place?

- Lines 157/158: Beirut as a type III-city according to Hahs et al. 2009: Hahs et al. refer to the year 1600 as a threshold (extensive transformations before or after 1600). As Beirut is a very old city, I guess that it might have experienced extensive transformations well before 1600. So, I am not convinced by your argument that Beirut is a type III-city – why not a type I-city? Please, be more precise in making your argument.

- Lines 180/181 University = American University of Beirut? Please, specify.

- Please, explain in more detail what TWINSPAN does, e.g. shortly mention what the cut levels are applied for. Also, please in the “Analysis” chapter, do explain how all this analysis is related to the target species. It is not mentioned at all how you identified habitat analogs for it.

Results:

- Line 207: What does “800 m” stand for?

- I think that in addition to (or instead of) Tables 3 and 5, ordination plots will be illustrative to show with which other species and with which life form types the target species preferentially occurs. The tables occupy a lot of space and are tedious to read.

- If you decide to stick with Tables 3 and 5, please define letters 1, 2, 3, … of the Domin-scale at the bottom of the tables.

- Line 244: add “as” between “around evergreen ornamentals such” and “the”.

- Line 246: Change “florsitics” to “floristics”

- Table 4: Three different codes (Life-form eight digit name, Abbreviated lifeform category, and Numeric code) are used here for the different life forms but no description/ definition is provided. Please, choose one out of three codes and add a description / definition of each life form type.

- Fig. 1: What exactly is shown here? Are these only the associates of M. crassifolia as stated in lines 263/264, i.e., species co-occurring with the target species, or are these all the species that you found across the 78 plots? As you worked with percent cover per species, it would as well be interesting to illustrate how the percent cover of the target species does change with the percent cover of specific life form types.

- Table 6: What does grey vs. no shade depict? What does “0” stand for?

- Lines 339/340: Rather “lowest representation” than “highest representation”? Otherwise, I do not understand how this is related to quadrat groups that excluded the target species.

- Table 8: Once, there is “chamaeophytes” written instead of “chamaephytes”. Please, correct.

Discussion:

- Parallel to the Introduction, the Discussion would benefit from better linking the study’s results to recent urban biodiversity research, especially to our knowledge about species conservation in urban areas, to the concept of novel ecosystems, and to functional traits and functional groups.

- Lines 365/366: “Such studies, however, are mostly conducted in natural habitats, and in many instances, deliberately exclude disturbed areas from sampling” – You do not compare natural to urban habitats, so why discuss this?

- Line 374: “aspect of urban vegetation that challenges field data analysis”: It might challenge classification but no analysis as such (see above). – same in line 384

- Line 387: How to develop an understanding of urban habitat analogs when there’s no description of natural habitats the target species usually occurs in (see above)?

- Lines 410-414: How to relate your method to LIDAR-data as long as herbaceous species cannot be distinguished by remote sensing (such as LIDAR)?

- Lines 422/423 “additional habitats”: You identified habitats where the target species occurs, habitats where it does not occur, and changes in percent cover of the target species across habitats. How did you identify additional habitats? Please explain in the Methods chapter.

- Lines 443: Add “of” between “case” and “established”.

- Lines 442-447: How would you manage a green space where several species of conservation interest are competing with each other? Would you selectively remove some of them?

6. PLOS authors have the option to publish the peer review history of their article (what does this mean?). If published, this will include your full peer review and any attached files.

Reviewer #1: Yes: Zerihun Woldu

Reviewer #2: No

Reviewer #3: No

---

## [Author Response · Author response to Decision Letter 0]

4 Mar 2020

We have performed a major revision to the manuscript as suggested by the reviewers. We thank the reviewers for their thorough and constructive feedback and we agree with all the feedback and suggested changes. Considering that major changes were incorporated to the text of the manuscript there may be differences in the revised manuscript presented in 'track changes' and the version with no 'track changes' as it became difficult to read through the track changes. All the modifications, however, are clearly stated in the response to reviewers letter. Thank you for this opportunity.

---

## [Decision Letter · Decision Letter 1]

3 Apr 2020

PONE-D-19-19705R1

Biodiversity conservation in cities: Defining habitat analogues for plant species of conservation interest

PLOS ONE

Dear Professor Talhouk,

Thank you for submitting your manuscript to PLOS ONE. After careful consideration, we feel that it has merit but does not fully meet PLOS ONE’s publication criteria as it currently stands. Therefore, we invite you to submit a revised version of the manuscript that addresses the points raised during the review process.

The rather minor comments provided by the two reviewers mainly refer to terminology and phrasing, as well as to some improvements how references, figures and tables are presented. Please consider all of the reviewers' comments thoroughly when revising your paper. I think these recommendations can easily be implemented, and I'm looking forward to receiving a revised version of your manuscript soon.

We would appreciate receiving your revised manuscript by May 18 2020 11:59PM. To enhance the reproducibility of your results, we recommend that if applicable you deposit your laboratory protocols in protocols.io, where a protocol can be assigned its own identifier (DOI) such that it can be cited independently in the future. For instructions see: http://journals.plos.org/plosone/s/submission-guidelines#loc-laboratory-protocols

We look forward to receiving your revised manuscript.

Kind regards,

Harald Auge

Academic Editor

PLOS ONE

Reviewers' comments:

Reviewer's Responses to Questions

**Comments to the Author**

1. If the authors have adequately addressed your comments raised in a previous round of review and you feel that this manuscript is now acceptable for publication, you may indicate that here to bypass the “Comments to the Author” section, enter your conflict of interest statement in the “Confidential to Editor” section, and submit your "Accept" recommendation.

Reviewer #2: All comments have been addressed

Reviewer #3: All comments have been addressed

2. Is the manuscript technically sound, and do the data support the conclusions?

Reviewer #2: Yes

Reviewer #3: Yes

3. Has the statistical analysis been performed appropriately and rigorously? 

Reviewer #2: Yes

Reviewer #3: Yes

4. Have the authors made all data underlying the findings in their manuscript fully available?

Reviewer #2: Yes

Reviewer #3: Yes

5. Is the manuscript presented in an intelligible fashion and written in standard English?

Reviewer #2: Yes

Reviewer #3: Yes

6. Review Comments to the Author

Reviewer #2: The authors have made a good effort in addressing the points I mentioned in the last review and I see that the manuscript has considerably improved. I have still a few suggestions that I think might be interesting to consider in order to further improve the manuscript.

ABSTRACT

There have been numerous and interesting improvements in the manuscript in the last review, but the abstract has barely changed. In particular, I think it might be interesting to include a short sentence about the new information provided on the local habitat conditions of M. crassifolia (i.e. the it occurs throughout a wide range of potential urban habitats).

INTRODUCTION

In the introduction, the authors have now made good use of the current literature in urban ecology. This section has now improved considerably.

METHODS

The methods are more concise and clear now. I also find the new figures on Beirut’s historical development and particularly on the location of the sites very informative.

RESULTS

Regarding the results section, I would move Table 1 to the Appendix, as I think it might be too long and the authors have already made a good summary of it in lines 340-343.

Please check if all the figures are properly numbered when uploading, as some of the attached figures didn’t match the numbering in the text.

I am happy to see that table 4 has information about the local habitat conditions of the sites.

DISCUSSION

The discussion has substantially improved as well. The authors now added clarifications regarding the important local habitat conditions of M. crassifolia and about the reasons why the plant might be missing in some sites.

GENERAL REMARKS

I think the manuscript has now improved considerably and can be suitable for publication after the aforementioned improvements.

Reviewer #3: Dear Itani et al.

Thank you for considering my previous comments. I think that the manuscript has greatly improved. Still, some minor issues remain. Please, allow me to point these out:

- Abstract, line 45: add “of” between “design” and “urban habitat analogues”

- Lines 72 to 74: This sentence is misleading. It sounds as if novel ecosystems do include remnants of natural vegetation. Please, have a look at Kowarik & von der Lippe (DOI: 10.1111/1365-2664.13144) for definitions of natural remnants vs. novel ecosystems and do rephrase the sentence.

- Lines 74/75: Isn’t management something intentional? So why “unintentionally managed”?

- Lines 76/77: "When unmanaged, urban green spaces … can potentially contribute to urban biodiversity conservation" – does that mean, on the contrary, that when managed, urban green spaces cannot contribute to urban biodiversity conservation? It sounds like this but I do not think that this is what you wanted to say, especially as in the end of your manuscript you suggest the intentional design and management of habitat analogues for the conservation of the target species. Please, rephrase lines 76/77.

- Please note that habitat = biotope (the former is in English and the latter is derived from German, but basically, both mean the same). So, I suggest you only use the term habitat, e.g. from lines 92 on: “Many plant species can be found more or less regularly in various urban habitats. For example, classification of urban habitat types inside the city of Berlin has revealed 19 habitats particularly worthy of protection and these were nominated as legally protected [22]. However, the classification of habitat types inside cities requires standard habitat classification systems which have not been developed in all countries, at least not in Lebanon [23]. Furthermore, the nomination of such habitat types becomes challenging when [...]”

- In lines 141-149, also use the terms habitat and classification: lines 145 “vegetation classification” instead of “vegetation description”; line 148 “vegetation classification” instead of “biotope type mapping”; lines 148 and 149 “habitats” instead of “biotopes”. The way the text is written at the moment makes me feel that different terms are used for the same thing, that’s why I am suggesting these replacements.

- Lines 159/160, I suggest you write “[…] other studies suggest that descriptions of functional types, such as life form, may permit […]”. At the moment, it reads as if functional types are an example of life form – but rather, life forms are an example of functional types.

- Line 163: I do not know what you mean with “other physiognomic characters”. Can you use another term or provide an example?

- Lines 197 to 203 are confusing. There, it seems that Khaldeh, Beirut (including Ras Beirut), Amchhit and Byblos are your study sites. However, in lines 215 ff, you write that Ras Beirut is your study site. Please, be more precise in lines 197 – 203.

- References to Fig. 1, 2, and 3 in the text are mixed up. Fig. 1 in the text = Fig. 2 where the figure is shown; similarly Fig. 2 in the text = Fig 3 and Fig 3 in the text is Fig.1.

- Line 263 “we placed quadrats” – a certain number? Similarly, lines 266/267 “we increased the number of quadrats” – up to a maximum number of …?

- Lines 304/305, the site numbers “(Site 17)” / “(Site 16)” do not seem to occur anywhere else in the manuscript (at least, I could not find them). I think that it is not necessary to provide these numbers as long as they are not referred to anywhere else (e.g. tables or figures). Therefore, I suggest to delete the numbers.

- Lines 447: “is classification is” – delete one “is”

- Lines 488/489: Shwartz et al. warn against expanding cities and they suggest to improve the quality of urban green spaces. But they do not present these two points as opposing strategies. Therefore, I suggest you phrase the sentence like this: “Improving the quality of existing green spaces throughout the entire urban matrix has been suggested as an effective approach to enhancing biodiversity experience [122].”

- Lines 550-552: This (trash) comes surprisingly and in my opinion has no close relationship to the core topic of your paper. I would delete it.

- Line 556: “increasing species’ site area” – do you mean all species or do you mean specific species (e.g. rare species, endemic species, protected species)?

- In Fig. 2 it is very hard to visually distinguish between dots for Status = Extinct and dots for Status = Recolonized. Please, consider changing colors so that differences among the colors become more obvious (or check in the proof if quality go better than it is in the pdf-file that I got for review).

- References: Please, do check references carefully as there are several typos within then. For example, journal names should be written with capital first letters (e.g. not Journal of applied ecology but Journal of Applied Ecology). Moreover, reference number 10 says “Doctoral dissertation” and “Master thesis” – it cannot be both a t once, can it? For reference number 19, rather cite the journal paper (Knapp, S., Kühn, I., Wittig, R., Ozinga, W.A., Poschlod, P. & Klotz, S. (2008) Urbanization causes shifts in species' trait state frequencies. Preslia 80, 375-388) than the book chapter, as the journal paper might be accessible to more readers. Also, in reference number 29, it should be “analogues” not “analogueues”. And with reference number 116, there’s an x after 116. There might be more typos as I did not look at them in detail.

7. PLOS authors have the option to publish the peer review history of their article (what does this mean?). If published, this will include your full peer review and any attached files.

Reviewer #2: No

Reviewer #3: No

---

## [Author Response · Author response to Decision Letter 1]

5 May 2020

PONE-D-19-19705

Biodiversity conservation in cities: Defining habitat analogs for plant species of conservation interest

PLOS ONE

RESPONSE TO REVIEWERS – SECOND FEEDBACK

Reviewer #2: The authors have made a good effort in addressing the points I mentioned in the last review and I see that the manuscript has considerably improved. I have still a few suggestions that I think might be interesting to consider in order to further improve the manuscript.

ABSTRACT

There have been numerous and interesting improvements in the manuscript in the last review, but the abstract has barely changed. In particular, I think it might be interesting to include a short sentence about the new information provided on the local habitat conditions of M. crassifolia (i.e. the it occurs throughout a wide range of potential urban habitats).

• As suggested by the reviewer we made major revisions to the abstract to reflect the revised version of the manuscript and to mention the new information on local habitat conditions of M. crassifolia

INTRODUCTION

In the introduction, the authors have now made good use of the current literature in urban ecology. This section has now improved considerably.

• We thank the reviewer for the positive feedback 

METHODS

The methods are more concise and clear now. I also find the new figures on Beirut’s historical development and particularly on the location of the sites very informative.

• We thank the reviewer for the positive feedback 

RESULTS

Regarding the results section, I would move Table 1 to the Appendix, as I think it might be too long and the authors have already made a good summary of it in lines 340-343.

• We moved Table 1 to the Appendix as suggested by the reviewer

Please check if all the figures are properly numbered when uploading, as some of the attached figures didn’t match the numbering in the text.

• We revised all figures and tables and made sure that their numbering is correct

I am happy to see that table 4 has information about the local habitat conditions of the sites.

• We thank the reviewer for his/her favorable comment

DISCUSSION

The discussion has substantially improved as well. The authors now added clarifications regarding the important local habitat conditions of M. crassifolia and about the reasons why the plant might be missing in some sites.

• We thank the reviewer for his/her favorable comment

GENERAL REMARKS

I think the manuscript has now improved considerably and can be suitable for publication after the aforementioned improvements.

Reviewer #3: Dear Itani et al.

Thank you for considering my previous comments. I think that the manuscript has greatly improved. Still, some minor issues remain. Please, allow me to point these out:

- Abstract, line 45: add “of” between “design” and “urban habitat analogues”

• We added “of” between “design” and “urban habitat analogues” as suggested by reviewer

- Lines 72 to 74: This sentence is misleading. It sounds as if novel ecosystems do include remnants of natural vegetation. Please, have a look at Kowarik & von der Lippe (DOI: 10.1111/1365-2664.13144) for definitions of natural remnants vs. novel ecosystems and do rephrase the sentence.

• The reference Kowarik and von der Lippe was consulted as suggested by reviewers and the sentence was revised for more clarification. “Novel ecosystems include urban green spaces that emerge mostly after built structures have replaced previously existing ecosystems. As such they include non native vegetation assemblages, consisting of native, spontaneous, naturalized, and invasive species [4]. ”

- Lines 74/75: Isn’t management something intentional? So why “unintentionally managed”?

• Sentence revised “Urban green spaces are sometimes abandoned after human disturbances or continue to experience disturbance regimes and consequently contain a range of both early- and late-succession vegetation. ”

- Lines 76/77: "When unmanaged, urban green spaces … can potentially contribute to urban biodiversity conservation" – does that mean, on the contrary, that when managed, urban green spaces cannot contribute to urban biodiversity conservation? It sounds like this but I do not think that this is what you wanted to say, especially as in the end of your manuscript you suggest the intentional design and management of habitat analogues for the conservation of the target species. Please, rephrase lines 76/77.

• Sentence in line 76/77 was rephrased to clarify that both unmanaged and managed green space can potentially contribute to urban biodiversity conservation. “Both unmanaged and managed green space can potentially contribute to urban biodiversity conservation.” added as introductory sentence.

- Please note that habitat = biotope (the former is in English and the latter is derived from German, but basically, both mean the same). So, I suggest you only use the term habitat, e.g. from lines 92 on: “Many plant species can be found more or less regularly in various urban habitats. For example, classification of urban habitat types inside the city of Berlin has revealed 19 habitats particularly worthy of protection and these were nominated as legally protected [22]. However, the classification of habitat types inside cities requires standard habitat classification systems which have not been developed in all countries, at least not in Lebanon [23]. Furthermore, the nomination of such habitat types becomes challenging when [...]”

• Habitat was used instead of biotope and sentence was changed as per reviewer suggestion

- In lines 141-149, also use the terms habitat and classification: lines 145 “vegetation classification” instead of “vegetation description”; line 148 “vegetation classification” instead of “biotope type mapping”; lines 148 and 149 “habitats” instead of “biotopes”. The way the text is written at the moment makes me feel that different terms are used for the same thing, that’s why I am suggesting these replacements.

• Terms were revised as suggested by reviewer

- Lines 159/160, I suggest you write “[…] other studies suggest that descriptions of functional types, such as life form, may permit […]”. At the moment, it reads as if functional types are an example of life form – but rather, life forms are an example of functional types.

• Sentence was revised as suggested by reviewer

- Line 163: I do not know what you mean with “other physiognomic characters”. Can you use another term or provide an example?

• We used another term as suggested by the reviewer “... life-form, among other descriptions of functional types, were associated with plant responses to environmental change...”

- Lines 197 to 203 are confusing. There, it seems that Khaldeh, Beirut (including Ras Beirut), Amchhit and Byblos are your study sites. However, in lines 215 ff, you write that Ras Beirut is your study site. Please, be more precise in lines 197 – 203.

• Lines 197 to 203 were revised to clarify that the study site was in Beirut

- References to Fig. 1, 2, and 3 in the text are mixed up. Fig. 1 in the text = Fig. 2 where the figure is shown; similarly Fig. 2 in the text = Fig 3 and Fig 3 in the text is Fig.1.

• References to figures were adjusted

- Line 263 “we placed quadrats” – a certain number? Similarly, lines 266/267 “we increased the number of quadrats” – up to a maximum number of …?

• Numbers were provided as suggested by the reviewer “In vegetation patches with clearly visible boundaries, one to two 1 m × 1 m quadrats were placed [32].” “... we increased the number of quadrats, up to six, … ”

- Lines 304/305, the site numbers “(Site 17)” / “(Site 16)” do not seem to occur anywhere else in the manuscript (at least, I could not find them). I think that it is not necessary to provide these numbers as long as they are not referred to anywhere else (e.g. tables or figures). Therefore, I suggest to delete the numbers.

• As suggested by reviewer sites 16 / 17 were deleted

- Lines 447: “is classification is” – delete one “is”

• Change made extra “is” removed

- Lines 488/489: Shwartz et al. warn against expanding cities and they suggest to improve the quality of urban green spaces. But they do not present these two points as opposing strategies. Therefore, I suggest you phrase the sentence like this: “Improving the quality of existing green spaces throughout the entire urban matrix has been suggested as an effective approach to enhancing biodiversity experience [122].”

• Sentence was rephrased as suggested by reviewer

- Lines 550-552: This (trash) comes surprisingly and in my opinion has no close relationship to the core topic of your paper. I would delete it.

• Sentence regarding trash deleted as suggested by reviewer

- Line 556: “increasing species’ site area” – do you mean all species or do you mean specific species (e.g. rare species, endemic species, protected species)?

• In this sentence we mean only the target species. The sentence was revised to clarify this. “... , increasing a target species’ site area in a city is highly desired ...”

- In Fig. 2 it is very hard to visually distinguish between dots for Status = Extinct and dots for Status = Recolonized. Please, consider changing colors so that differences among the colors become more obvious (or check in the proof if quality go better than it is in the pdf-file that I got for review).

• Color scheme was changed to better appear after publication

- References: Please, do check references carefully as there are several typos within then. For example, journal names should be written with capital first letters (e.g. not Journal of applied ecology but Journal of Applied Ecology). Moreover, reference number 10 says “Doctoral dissertation” and “Master thesis” – it cannot be both a t once, can it? For reference number 19, rather cite the journal paper (Knapp, S., Kühn, I., Wittig, R., Ozinga, W.A., Poschlod, P. & Klotz, S. (2008) Urbanization causes shifts in species' trait state frequencies. Preslia 80, 375-388) than the book chapter, as the journal paper might be accessible to more readers. Also, in reference number 29, it should be “analogues” not “analogueues”. And with reference number 116, there’s an x after 116. There might be more typos as I did not look at them in detail.

• We checked and corrected all references.

---

## [Editor Report · Decision Letter 2]

21 May 2020

Biodiversity conservation in cities: Defining habitat analogues for plant species of conservation interest

PONE-D-19-19705R2

Dear Dr. Talhouk,

We are pleased to inform you that your manuscript has been judged scientifically suitable for publication and will be formally accepted for publication once it complies with all outstanding technical requirements.

With kind regards,

Harald Auge

Academic Editor

PLOS ONE
---

## [Editor Report · Acceptance letter]

28 May 2020

PONE-D-19-19705R2 

Biodiversity conservation in cities: Defining habitat analogues for plant species of conservation interest 

Dear Dr. Talhouk:

I am pleased to inform you that your manuscript has been deemed suitable for publication in PLOS ONE. Congratulations! Your manuscript is now with our production department. 

With kind regards,

on behalf of

Dr. Harald Auge 

Academic Editor

PLOS ONE